# Capicua regulates neural stem cell proliferation and lineage specification through control of Ets factors

Shiekh Tanveer Ahmad [1,2,3], Alexandra D. Rogers[2], Myra J. Chen [2,3], Rajiv Dixit[2,3,4], Lata Adnani[3,4], Luke S. Frankiw[2,5], Samuel O. Lawn[2,6], Michael D. Blough[2], Mana Alshehri[2,7], Wei Wu[2,8], Marco A. Marra [9], Stephen M. Robbins[2], J. Gregory Cairncross[2], Carol Schuurmans[3,4] & Jennifer A. Chan[1,2,3]

Capicua (Cic) is a transcriptional repressor mutated in the brain cancer oligodendroglioma. Despite its cancer link, little is known of Cic's function in the brain. We show that nuclear Cic expression is strongest in astrocytes and neurons but weaker in stem cells and oligodendroglial lineage cells. Using a new conditional *Cic* knockout mouse, we demonstrate that forebrain-specific *Cic* deletion increases proliferation and self-renewal of neural stem cells. Furthermore, *Cic* loss biases neural stem cells toward glial lineage selection, expanding the pool of oligodendrocyte precursor cells (OPCs). These proliferation and lineage effects are dependent on de-repression of Ets transcription factors. In patient-derived oligodendroglioma cells, CIC re-expression or ETV5 blockade decreases lineage bias, proliferation, self-renewal, and tumorigenicity. Our results identify Cic as an important regulator of cell fate in neurodevelopment and oligodendroglioma, and suggest that its loss contributes to oligodendroglioma by promoting proliferation and an OPC-like identity via Ets overactivity.

[1] Department of Pathology & Laboratory Medicine, University of Calgary, Calgary, AB, Canada. [2] Charbonneau Cancer Institute, University of Calgary, Calgary, AB, Canada. [3] Alberta Children's Hospital Research Institute, University of Calgary, Calgary, AB, Canada. [4] Sunnybrook Research Institute and the Department of Biochemistry, University of Toronto, Toronto, ON, Canada. [5] Department of Biology & Bioengineering, California Institute of Technology, Pasadena, CA, USA. [6] Zymeworks, Vancouver, BC, Canada. [7] King Abdullah International Medical Centre, Riyadh, Saudi Arabia. [8] University of California, San Francisco, CA, USA. [9] BC Genome Sciences Centre, Vancouver, BC, Canada. Correspondence and requests for materials should be addressed to J.A.C. (email: jawchan@ucalgary.ca)

The identification of cancer genes often presents opportunities to uncover previously unappreciated mechanisms regulating development. The transcriptional repressor Capicua (CIC) has been identified as a likely tumor suppressor, as CIC mutations and/or reduced expression are found in several cancers. In the brain, CIC mutations are nearly exclusively found in oligodendrogliomas (ODGs)—tumors composed of cells resembling oligodendrocyte precursor cells[1,2]. Indeed, concurrent IDH1/2 mutation, single-copy losses of 1p and 19q, and mutation of the remaining copy of CIC on chr 19q13 are together highly characteristic of ODG[3–5]. These associations suggest a unique relationship between CIC and glial biology.

Prior work has shown that Cic is a transcriptional repressor downstream of receptor tyrosine kinase (RTK) signaling[6]. Binding of Cic to the sequence T(G/C)AATG(G/A)A in enhancers and promoters leads to transcriptional repression of its target genes[7,8]. This default repression is relieved upon RTK signaling[6,9–11], permitting transcription of targets—among which are PEA3/ETS transcription factors ETV1/4/5[12]. To date, knowledge of Cic's function in mammalian development or in the brain is limited. Yang et al. using Cic conditional knockout mice, reported that Cic loss increases a population of proliferating Olig2 + cells in the brain, and potentiates tumorigenesis in a PDGFR-driven glioma model[13]. The mechanisms underlying those findings, however, remained undefined. Meanwhile, in the non-neoplastic context, Lu et al.[14] showed that impairing the interaction of Cic with Ataxin1 results in neurobehavioral and neurocognitive phenotypes, and alteration in cortical neuronal populations—indicating a role for Cic in neuronal biology as well. Deciphering the molecular mechanisms of Cic function may shed light on these varied phenotypes.

Here, we examine Cic expression in the mammalian forebrain, and take a loss-of-function approach to determine its role in specifying neuronal-glial identity. Our results reveal an important role for Cic in regulating the proliferation and lineage specification of neural stem cells—with loss favoring neural stem cell (NSC) proliferation and glial production at the expense of neuronal production. Furthermore, we show that these effects are mediated largely through Cic's regulation of Ets factors. The proliferative dysregulation is recapitulated in ODG cells, where CIC re-expression or Ets blockade reduces tumorigenicity. Our findings reveal an important role for CIC in development and ODG, and identify Etv5 as a potential therapeutic target for ODG.

## Results

### Nuclear Cic levels are cell type- and stage-specific.
To investigate Cic's potential function in forebrain development and ODG, we first examined its expression in several regions and cell types in the mouse brain. Immunofluorescence staining revealed that all cell types examined have some detectable level of Cic, whether cytoplasmic or nuclear. Because of its previously identified role as a transcriptional repressor, however, we quantitated nuclear Cic staining intensity across cell types and stages.

Focusing first on the NSC compartment, we assessed tissue from the dorsal telencephalon at embryonic day (E) E12. At this stage, non-lineage restricted NSCs are found in the ventricular zone (VZ) where they can be identified by their location and expression of the transcription factor Sox2[15]. Consistent with the previous demonstration that RTK signaling is locally elevated in the embryonic VZ[16,17] and that Cic nuclear export is regulated by ERK-mediated Cic phosphorylation[6,9–11], Sox2 + VZ cells showed Cic localization that was predominantly cytoplasmic, with weaker nuclear expression (Fig. 1a). Pools of stem/progenitor cells are not limited to development, but also persist postnatally in the subventricular zone (SVZ) of the lateral

ventricle and in the subgranular layer (SGL) of the hippocampal dentate gyrus[18,19]. In the SVZ and SGL of postnatal day (P) P21 and P56 mice, nuclear Cic was also weak in Sox2 + cells—in contrast to stronger nuclear expression in adjacent differentiated cells (Fig. 1b, c). Thus, in embryonic and postnatal brain, NSCs have low levels of nuclear Cic.

As cells differentiated, Cic nuclear localization increased, but with notable differences between cell types. In P56 cortex, NeuN + neurons showed the strongest nuclear Cic (Fig. 1i, j). The increase in Cic within the neuronal lineage was detectable during embryonic neurogenesis, with a modest increase detected as cells transitioned from Sox2 + stem cells to Tbr2 + neuronal intermediate progenitors, and then a more-marked elevation in Tbr1 + post-mitotic neurons in the intermediate zone and cortical plate, where nuclear Cic levels approached those of adult cortical neurons (Fig. 1g–j). Nuclear Cic was also increased in GFAP + or Aldh1 + astrocytes in the cortex and white matter relative to Sox2 + cells (Fig. 1d; Supplementary Fig. 1a). The lowest levels of nuclear Cic were found in the oligodendroglial lineage. Olig2 is a bHLH transcription factor that is expressed in cells at or before oligodendrocyte specification and continues to be present throughout oligodendroglial differentiation[20,21]. Olig2 + Pdgfra + OPCs had the lowest levels of all, with a modest increase in CNPase + immature oligodendrocytes, and a further increase in mature CC1 + oligodendrocytes (Fig. 1e, f, j; Supplementary Fig. 1b). Overall, the mean nuclear signal intensity for Cic was significantly lower in Sox2 + and Pdgfra + cells than in NeuN + cells, Tbr1 + cells, or Gfap + cells (Fig. 1j). As both NSCs and OPCs are proliferation-competent cell types, this pattern of expression raised the possibility that CIC may repress proliferation-related genes and/or early oligodendroglial-promoting programs.

### Cic loss increases glial cells at the expense of neurons.
Domains in Cic include an HMG box and a C-terminal C1 domain that together mediate DNA binding, and a C-terminal Gro-L domain that mediates protein–protein interactions[10,22–25]. We generated Cic conditional knockout mice in which exons 2–11 of Cic were flanked by loxP sites, with the floxed region containing all exons encoding the HMG box. Upon Cre expression, exons 2–11 are excised and the remaining exons 12–20 are frameshifted (Fig. 2a), ablating all of these critical domains. We used these animals for in vivo studies and for cell line generation to dissect Cic's functions.

We crossed Cic-floxed mice to Foxg1-Cre mice[26], to generate forebrain-specific Cic deletion starting at E10.5 (Fig. 2a, b). $Cic^{Fl/Fl};FoxG1^{Cre/+}$ animals were born in approximate Mendelian ratios and were grossly normal at birth, but became visible runts by P7, and were lethal by P22. The reason for lethality is unclear, but we suspect that poor feeding secondary to impaired neurologic function may be related to their decline. Although all major forebrain structures (e.g., cortex, white matter, deep nuclei, hippocampi) were present, and the cortex was laminated; Cic-null cerebra were smaller than those of littermate controls (Fig. 2c). The smaller cerebral size was due to global decreases in gray matter and white matter (Supplementary Fig. 2a–c). Despite decreased thickness/size of corpus callosum and cortex, however, the tissue showed increased cellularity (Supplementary Fig. 2b–d). As neuronal density was not significantly different in the knockouts, (Supplementary Fig. 2e), the findings suggested a decrease in total neurons and an increase in glia.

Evaluation of P21 cortices confirmed shifted cell proportions in Cic-null brains. NeuN + cells were decreased in Cic-null cortices compared with Cic-wildtype or heterozygote control cortices (Fig. 2d, e). In contrast, Gfap + cells were increased in knockouts relative to controls (Fig. 2d, e). Olig2 + cells were also

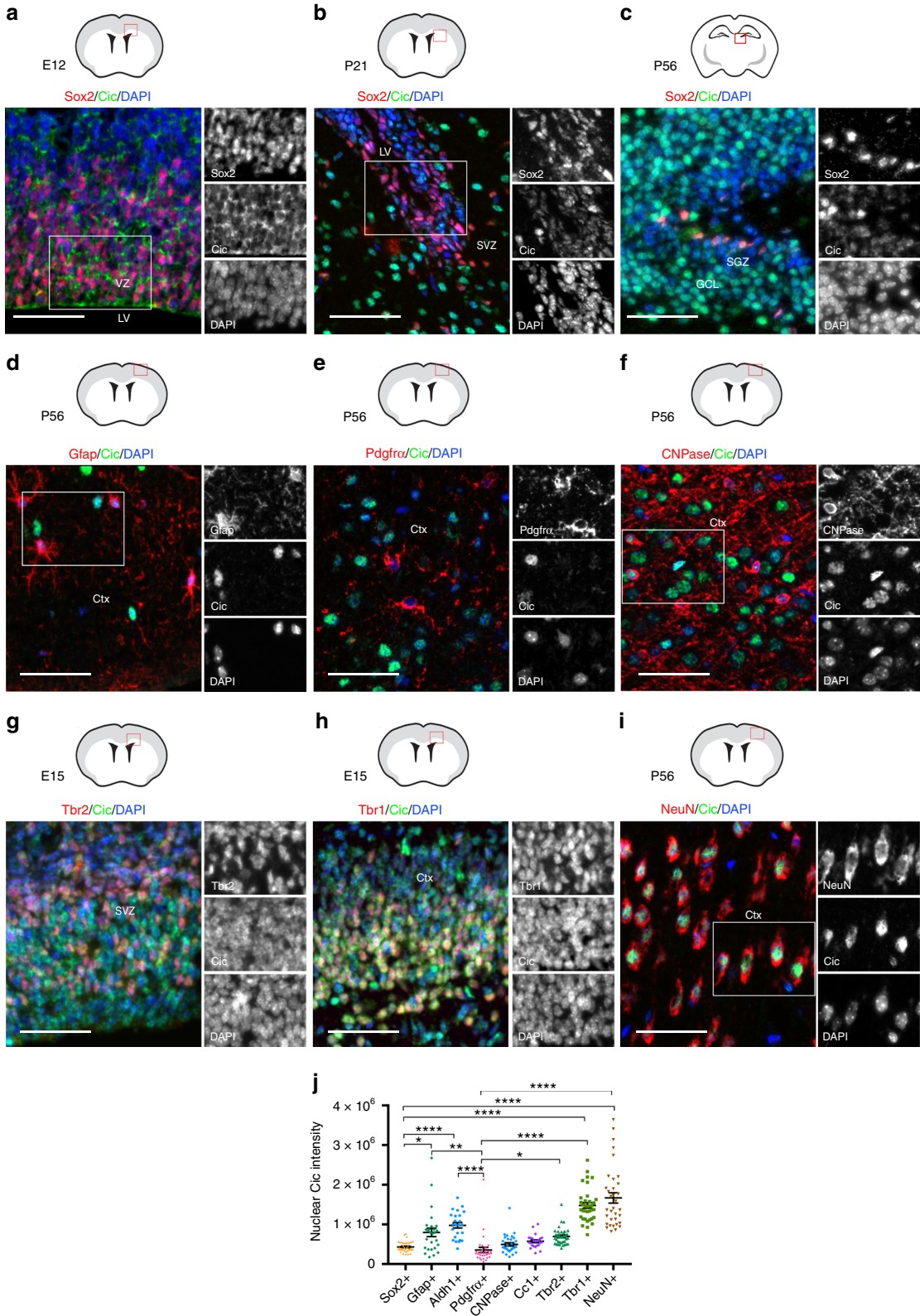

significantly increased (Fig. 2f, g)—a finding corroborated by increased Sox10 + (Fig. 2d, e). In oligodendroglial lineage progression, earlier stages were particularly increased, whereas later stages were decreased in *Cic*-null animals. Olig2 + Sox2 + cells and Olig2 + Pdgfra + OPCs were increased (Fig. 2f, g and Supplementary Fig. 3), whereas CNPase + immature oligodendrocytes were decreased (Fig. 2f, g) in *Cic*-null animals. Mbp,

the product of mature oligodendrocytes, was also decreased in *Cic*-null brains (Fig. 2f, g).

We found no change in apoptosis in the knockout brains (Supplementary Fig. 4f), suggesting that the phenotype of altered cell proportions was not due to increased cell death. Other possibilities were that Cic loss could be affecting NSC lineage selection, or that Cic loss could have specific effects on OPCs.

**Fig. 1** Differential Cic expression among cell types in the developing and mature brain. **a–h** Representative images of immunofluorescence staining for Cic expression in **a** Sox2 + stem cell populations in E12 subventricular zone, **b** P21 subventricular zone, and **c** P56 hippocampal dentate gyrus subgranular zone. Cic expression in glial cells in the adult brain showing localization in **d** Gfap + cortical astrocytes, **e** Pdgfra + white matter OPCs, and **f** CNPase + mature oligodendrocytes. Cic expression in neuronal populations in **g** E15 Tbr2 + early-born neurons in the SVZ and IZ, **h** E15 Tbr1 + late-born neurons in the cortical plate, and **i** adult post-mitotic NeuN + cortical neurons. **j** Quantitation of Cic nuclear staining intensities, plotted by brain cell types. All quantitation performed on P56 brain except Tbr2 and Tbr1, which were analyzed at E15. Each point represents one cell quantitated. Results from a minimum of 24 cells per marker from the boxed regions used for image analysis; data are from $n = 5$ brains. Pairwise comparisons between cell types performed by ANOVA with Tukey's post hoc test. Source data are provided as a Source Data file. Scale bar: 50 μm. Data shown as mean ± SD. *$p < 0.05$, **$p < 0.01$, ****$p < 0.0001$. Labels: LV–lateral ventricle, GCL–granule cell layer, SGZ–subgranular zone, SVZ–subventricular zone, Ctx–cortex

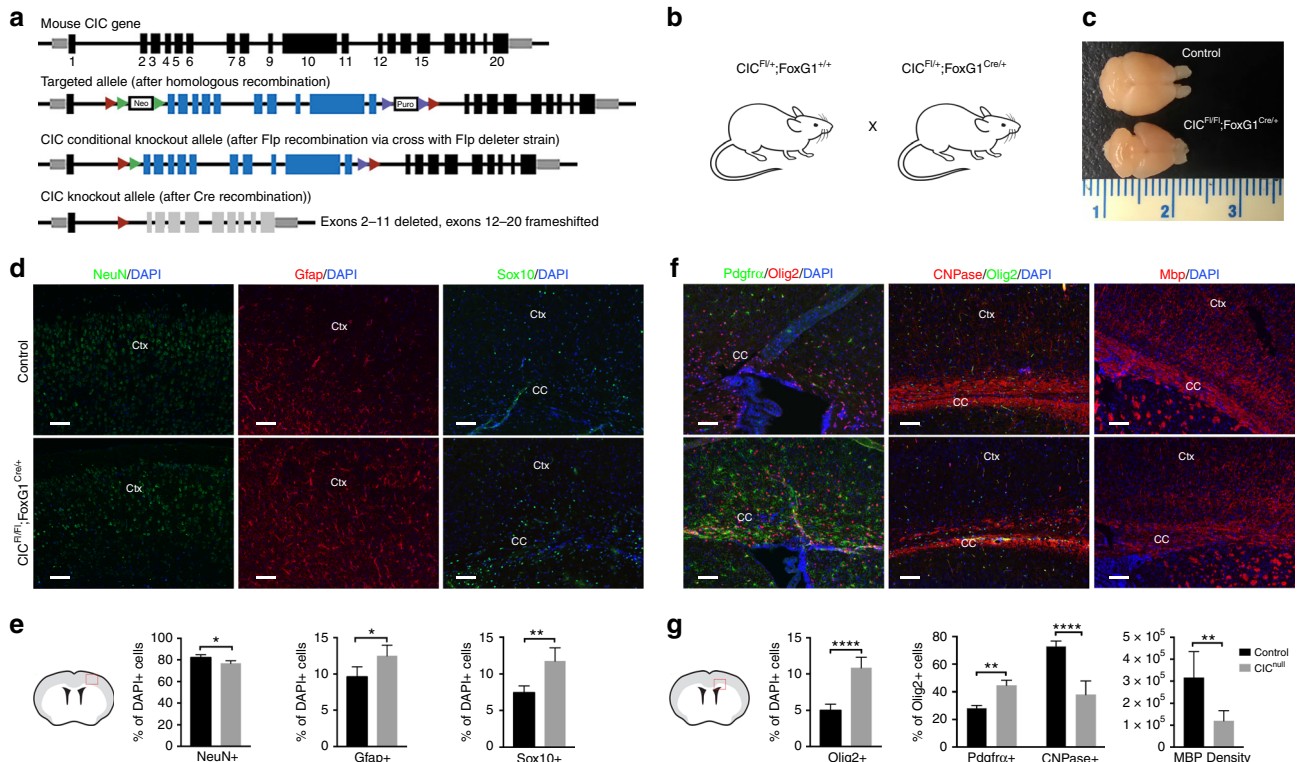

**Fig. 2** Forebrain-specific *Cic* deletion increases glial cells at the expense of neurons. **a** Targeting strategy for Cic conditional knockout mice. Exon numbering is shown relative to Cic transcript variant 1. **b** Forebrain-deletion of Cic starting from E10.5 by crossing CIC-floxed line with FoxG1-cre. *Cic^{Fl/Fl};FoxG1^{Cre/+}* animals are compared with *Cic^{Fl/+};FoxG^{Cre/+}* or *Cic^{Fl/Fl};FoxG1^{+/+}* as controls. **c** Representative gross morphology of *Cic*-deleted and *Cic*-wildtype brains at P21. **d, e** Representative staining **d** and total quantitation **e** of NeuN + , Gfap + , and Sox10 + cells in Cic-deleted (*Cic^{Fl/Fl};FoxG1^{Cre/+}*) cortex vs control (*Cic^{Fl/Fl};FoxG1^{+/+}*). Scale bar: 50 μm. **f, g** Representative staining **f** and total quantitation **g** of Olig2 + , Pdgfra + , and CNPase + cells in lateral corpus callosum, and Mbp expression in lateral corpus callosum at P2. Data from $n = 5$ mice per each group with the exception of Pdgfra staining ($n = 3$ control, $n = 4$ Cic-null) and Mbp staining ($n = 6$). Statistical analyses performed by unpaired $t$ test. Scale bar: 50 μm. Source data are provided as a Source Data file. Data shown as mean ± SD. *$p < 0.05$, **$p < 0.01$, ****$p < 0.0001$. Labels: Ctx–cortex, CC–corpus callosum

In the following work, we focus our investigations on Cic's role at the NSC stage, and examine its role in proliferation and lineage selection.

**Cic deficiency increases NSC proliferation and self-renewal.** To determine whether Cic loss affects NSC proliferation, we electroporated pCIG2-Cre (or pCIG2 empty control) into E13 *Cic^{Fl/Fl}* embryos and performed EdU labeling in the last 30 min prior to sacrifice. Forty-eight hours post electroporation, the fraction of GFP + cells that was EdU + was markedly increased in cre- vs. control-electroporated brains (Fig. 3a, b). These findings supported a cell-autonomous increase in NSC proliferation with CIC loss. There was also an increase in EdU + cells among non-GFP cells in the electroporated areas, suggesting additional non-cell-autonomous effects that we did not pursue (Supplementary Fig. 6e). To confirm the cell-autonomous gains in NSC proliferation, we turned to cell culture. *Cic*-null and control NSCs

were derived from E15 *Cic*-floxed cells transfected ex vivo with Cre or control plasmids (Fig. 3c). 3 days after plating equal cell numbers in NSC proliferation media, however, we found a threefold increase in numbers of Cic-null cells relative to controls (Fig. 3d). These data were corroborated by Alamar blue assay (Supplementary Fig. 4a), Ki67 immunostaining (Fig. 3e) and cell cycle analysis (Fig. 3f), which showed increased viability and cycling in Cic-null cells. There were no differences in the numbers of dead cells between the two (Supplementary Fig. 4c). Together, the data show that Cic is a strong negative regulator of proliferation in forebrain NSCs.

Self-renewal of *Cic*-null cells was also assessed by clonogenic assay (Fig. 3g). In this assay, after plating equal numbers of dissociated cells in semi-solid media, a higher number of spheres indicates higher number of self-renewing cells in the initial population, whereas sphere volume is a more general indicator of proliferation that includes effects from cell cycle kinetics, modes

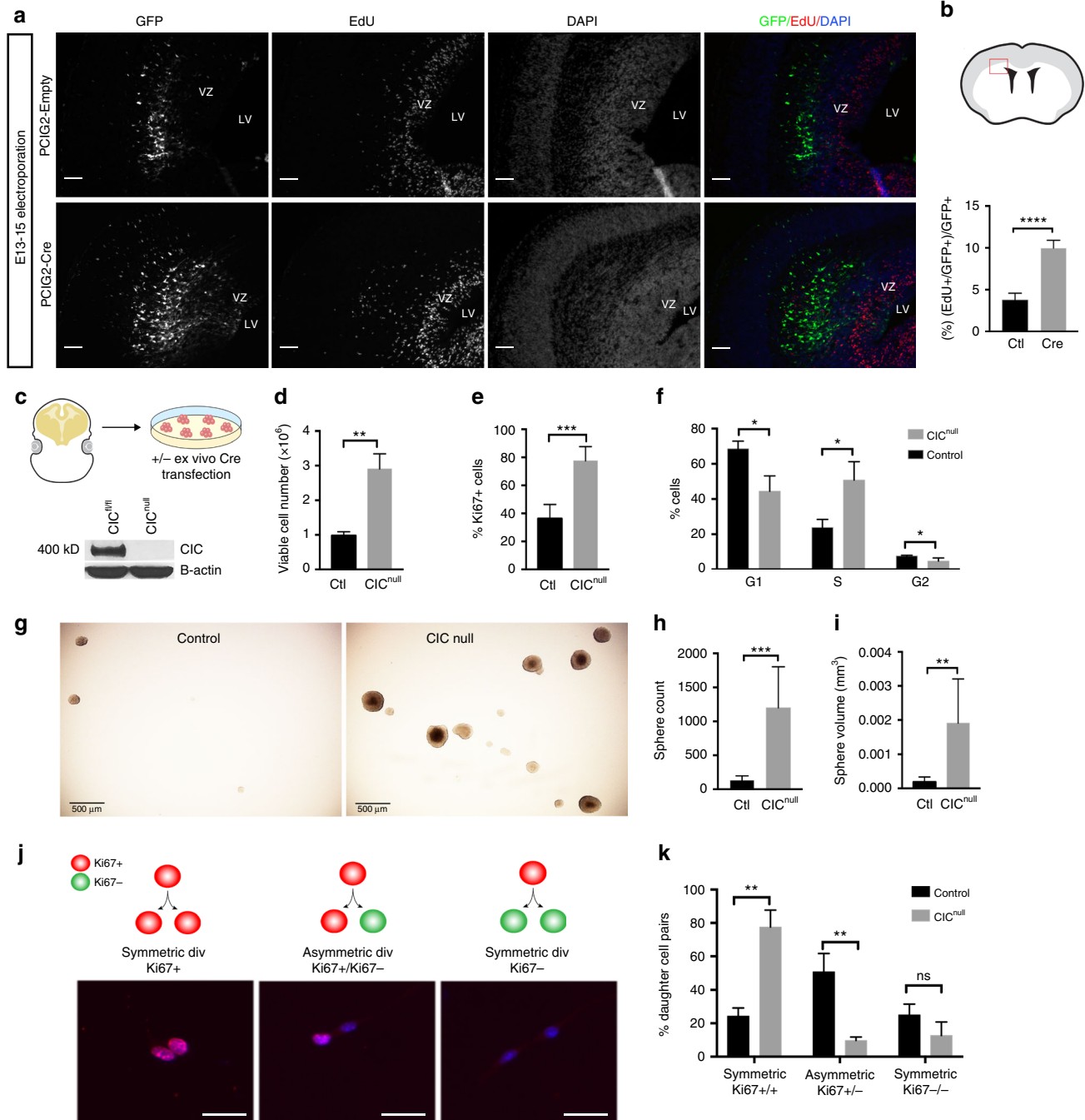

**Fig. 3** *Cic* deficiency increases proliferation and self-renewal of neural stem/progenitor cells. **a, b** EdU incorporation 48-hours after electroporation of *Cre* or control plasmid into VZ of E13 *Cic*-floxed embryos; **a** representative images of staining and **b** quantitation in boxed zone; $n = 4$ for pCig-cre, $n = 5$ for control; scale bar: 50 μm. **c** Generation of Cic-null cells and control cells from *Cic*-floxed cells via ex vivo transfection of Cre recombinase, and western blotting for validation of knockout. **d–f** Cell proliferation measured by Trypan blue assay **d**, Ki67 proliferation index **e**, and propidium iodide cell cycle analysis **f** in cultured cells; $n = 3$ biologic replicates. **g–i** Neural colony-forming assay, representative images **g** and quantitation of sphere number **h** and sphere size **i** from $n = 3$ biologic replicates; scale bar: 500 μm. **j–k** Paired cell assay schematic and representative images of Ki67 immunostaining **j**, and quantitation of symmetric proliferative, asymmetric, and symmetric terminal divisions **k**; data from $n = 3$ biologic replicates per condition; scale bar: 20 μm. Source data are provided as a Source Data file. Data shown as mean ± SD. ns–not significant, $*p < 0.05$, $**p < 0.01$, $***p < 0.001$, $****p < 0.0001$. Labels: VZ–ventricular zone, LV–lateral ventricle

of cell division, and the fraction of cells remaining in or exiting the cell cycle. Both *Cic*-null and control NSCs generated spheres, but *Cic*-null NSCs generated more spheres (Fig. 3g, h) and larger spheres (Fig. 3g, i) compared with controls. Thus, *Cic* loss confers not only higher proliferation but higher self-renewal in NSCs, at least when cells are in conditions promoting NSC proliferation.

**Cic regulates NSC cell division asymmetry.** During neurodevelopment, the NSC pool is first expanded by symmetric proliferative divisions, followed by rounds of asymmetric divisions of stem and progenitor cells generating neurons and glia, and finally by terminal symmetric differentiative divisions[18]. The balance between symmetric and asymmetric divisions is important;

deficiency in asymmetric divisions and an increase in symmetric divisions can lead to neoplasia[27].

To investigate whether alteration of cell division mode was a feature of the *Cic*-deficient NSCs, we performed paired cell assays. *Cic*-null and -control NSCs were seeded at low density in adherent cultures for 24 h, then fixed and stained for Ki67 (Fig. 3j). Pairs where both daughter cells were Ki67 + were scored as symmetric proliferative. Pairs where daughter cells differed in Ki67 (i.e., Ki67+/Ki67−) were scored as asymmetric. Pairs where both daughter cells were Ki67− were scored as symmetric terminal. *Cic*-null cells underwent more frequent symmetric proliferative divisions and fewer asymmetric divisions compared with controls (Fig. 3k). Although not significant, there was a trend to decreased symmetric terminal divisions between *Cic*-null and control cells (Fig. 3k). The net effect of these alterations in cell division symmetry/asymmetry is the presence of more cycling and self-renewing cells.

Consistent with the above increased NSC proliferation and self-renewal observed in vitro, examination of our electroporated brains showed that the fraction of GFP + cells that were Sox2 + was higher in cre- versus control-electroporated brains (Fig. 4a–c). There was no change in activated Caspase-3 (Supplementary Fig. 4d, e), indicating that the increased Sox2 + fraction in vivo was not owing to increased apoptosis of other cells. Cumulatively, the findings of increased NSC proliferation, decreased asymmetric divisions, and increased Sox2 + cells suggested that Cic deficiency increases a population of self-renewing (albeit dysregulated) stem-like cells.

**Gliogenic determinants are elevated in Cic-deficient NSCs.** Although the proliferative effects of Cic deficiency in NSCs were

clear, what was intriguing was the added possibility that Cic loss may be affecting lineage selection. As the *Cic^{Fl/Fl};FoxG1^{Cre/+}* phenotype was characterized by fewer neurons and increased glia, we asked whether Cic loss altered the expression of glial lineage selection factors and, thus, gliogenic potential.

To this end, we examined Sox9 and Olig2 expression in the NSCs and their progeny after E13 Cre- or control-electroporation. Sox9 is an HMG box transcription factor present in a range of central nervous system cells, including stem cells, glial precursors, and later glia. It has key roles in stem cell maintenance and in driving differentiation programs away from neurogenesis, toward gliogenesis, at the stage of gliogenic initiation[28–30]. Similarly, Olig2 is expressed in many cells including stem cells, oligodendroglial lineage cells, and some neurons; but it has a major role in establishing oligodendroglial competence. Two days post electroporation, the fraction of GFP + cells expressing Sox9 was significantly increased over controls (Fig. 4d, e). Five days post electroporation, a smaller fraction of *Cic*-deleted NSCs became Tbr1 + early-born (Fig. 4f, g). Conversely, greater fractions became Aldh1 + astrocytes and Pdgfra + OPCs (Fig. 4h–l).

These findings were echoed in the expression of stem and lineage markers in cultured NSCs. When maintained in serum-free NSC proliferation media, both *Cic*-null and control NSCs strongly expressed Nestin, as expected, and were devoid of staining for Gfap, bIII-Tubulin (as detected by Tuj1), and committed oligodendroglial markers (Fig. 5a–c; Supplementary Fig. 3c). There, was, however, a marked increase in the percentage of cells expressing Sox9 and Olig2 in *Cic*-null cultures (Fig. 5a–c). It is reasonable to posit that the increased Sox9 and Olig2 present in *Cic*-null NSCs then sets an intrinsic foundation for pro-glial

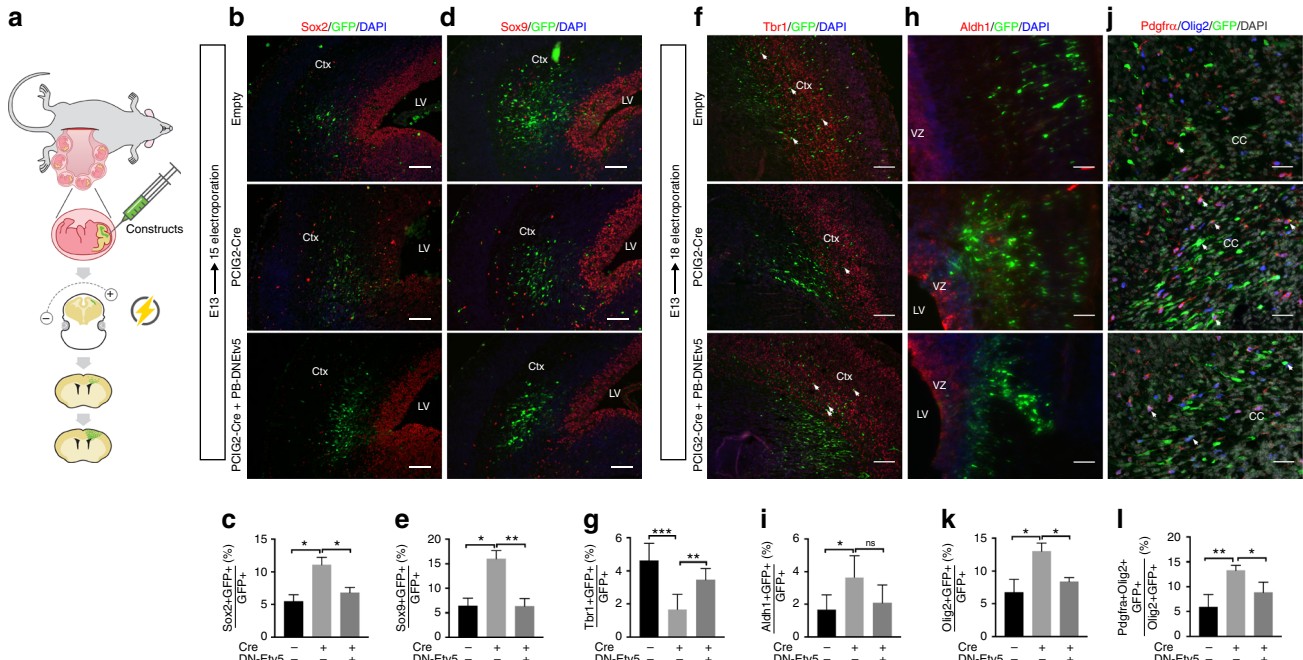

**Fig. 4** Deletion of *Cic* in the neurogenic period biases NSCs to glial lineage selection. **a** Localized deletion of *Cic* in neural stem cells via in utero electroporation into VZ cell of *Cic*-floxed embryos; targeted cells carry a fluorescent marker and express either cre recombinase (Cre), dominant negative ETV5 (DNETV5), or not (Empty control), depending on the electroporated plasmid. **b–l** Immunofluorescence staining and quantitation of cells in the electroporated patch. Sox2 + stem cells **b**, **c** and Sox9 + glioblasts **d**, **e** in n = 4 per group 2 days after E13 electroporation, analyzed at E15. Late-born Tbr1 + neurons **f**, **g**, Aldh1 + astrocytes **h**, **i**, Olig2 + oligodendrogial lineage cells **j**, **k**, and Pdgfra + oligodendrocyte precursors cells **l** detected 5 days after E13 electroporation, analysis at E18. Data from n = 5 for control and n = 6 mice for Cre for Tbr2 staining; n = 6 for control and n = 5 for cre for Aldh1 staining; and n = 4 per group for Olig2 and Pdgfra staining. Statistical analyses between control and experimental groups performed by ANOVA with Tukey's post hoc test. Scale bars: 50 μm. Source data are provided as a Source Data file. Data shown as mean ± SD. *p < 0.05, **p < 0.01, ***p < 0.001, ns–not significant. Labels: Ctx–cortex, LV–lateral ventricle, VZ–ventricular zone

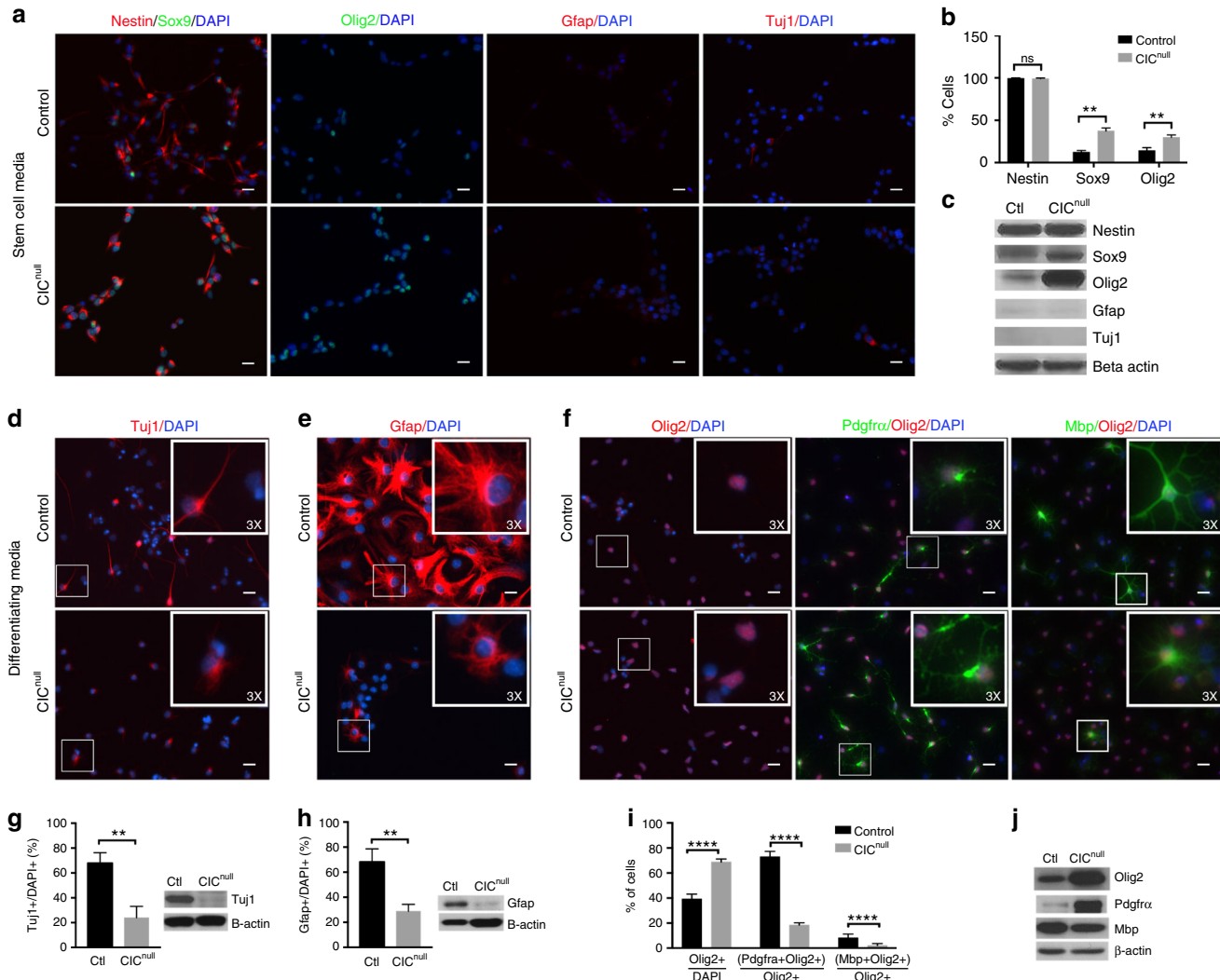

**Fig. 5** Cultured *Cic*-null cells are oligodendrocyte lineage-biased. **a–c** Cic-null and control mouse NSCs, cultured in serum-free stem cell proliferation media and analyzed for expression of Nestin, Sox9, Olig2, Gfap, and bIII-Tubulin (Tuj1). Representative images of immunofluorescence staining **a**; quantitation of cell percentages expressing Nestin, Sox9, and Olig2 **b**; and western blots for Nestin, Sox9, Olig2, Gfap, and Tuj1, with b-Actin loading control **c**. **d–j** Responses of Cic-null and control NSCs to 10-day exposure to lineage-directed differentiation conditions for neurons, astrocytes, and oligodendrocytes. Representative staining and quantitation of Tuj1 + cells **d**, **g** in cultures exposed to neuronal-promoting conditions. Representative staining and quantitation of Gfap + cells **e**, **h** in cultures exposed to astrocyte-promoting conditions. Representative staining and quantitation of Olig2 + cells as a percentage all cells, Pdgfra + Olig2 + cells as a percentage of Olig2 + cells, and Mbp + Olig2 + cells as a percentage of Olig2 + clls **f**, **i** in cultures exposed to oligodendrocyte-promoting conditions, with western blotting **j** for Olig2, Pdgfra, and Mbp. Data from $n = 3$ biologic replicates per condition for all experiments. Source data are provided as a Source Data file. Statistical analysis performed by unpaired $t$ test. Scale bar: 10 µm. Data shown as mean ± SD. ns–not significant, **$p < 0.01$, ****$p < 0.0001$

or pro-oligodendroglial programs starting early in the neural cellular hierarchy.

**Cic-deficient NSCs are biased to the oligodendroglial lineage.** To directly test cell type specification capacity, we challenged *Cic*-null and control NSCs with exposure to different lineage-promoting culture conditions. Neuronal differentiation was induced by culturing cells with B27 and cAMP. Astrocytic differentiation was induced by culturing NSCs in 1% FBS and $N_2$. Oligodendroglial differentiation was induced by culturing cells in media with B27 and tri-iodo-thyronine. After 10-day exposures to these conditions, cultures were analyzed for cellular identity and morphology.

Following exposure to neuronal-promoting conditions, Tuj1 + cells comprised the majority of cells in *Cic*-wildtype cultures,

but were only a minority in *Cic*-null cultures (Fig. 5d, g). Furthermore, the Tuj1 + *Cic*-null cells that were present had fewer and less complex cell processes than in their control counterparts (Fig. 5d). Likewise, in response to astrocyte-promoting conditions, Gfap + cells comprised the majority of cells in control cultures, but only a minority in *Cic*-null cultures (Fig. 5e, h); and the Gfap + *Cic*-null cells that were present displayed more rudimentary processes than controls (Fig. 5e, h). In the oligodendrocyte-promoting condition, there was a higher fraction of Olig2 + cells in *Cic*-null cultures compared with control, as well as increased Olig2 by western blotting (Fig. 5f–j). When analyzed for markers of OPCs versus more mature oligodendrocytes, *Cic*-null cultures also had a greater percentage of Olig2 + Pdgfra + cells than *Cic*-wildtype cultures (Fig. 5f–j). Conversely, the percentage of Olig2 + MBP + cells was decreased in *Cic*-null cultures (Fig. 5f–j). Furthermore, in

the few Olig2 + MBP + cells that were present in *Cic*-null cultures, process formation was rudimentary compared with *Cic*-wildtype cells (Fig. 5f). Thus, *Cic* is required for appropriate NSC responsiveness to extrinsic specification/differentiation cues. Our findings indicate that Cic-deficient NSCs have more limited ability to specify neuronal or astrocytic fates, but have increased permissiveness to the oligodendroglial lineage. Some findings suggest an additional maturation defect after lineage commitment, but we did not address this here.

We then asked what *Cic*-null NSCs became in the astrocytic and neuronal conditions, if not astrocytes and neurons. In the astrocyte-promoting condition, the Gfap- population was comprised predominantly of Sox2 + cells and, to a lesser extent Olig2 + cells, both of which were significantly increased in *Cic*-null cultures (Fig. 6a, b). Tuj1 + cells were rare in this condition, and did not differ between *Cic*-null cells and controls. In the neuronal-promoting condition, the Tuj1- population was similarly comprised predominantly of Sox2 + cells and Olig2 + cells

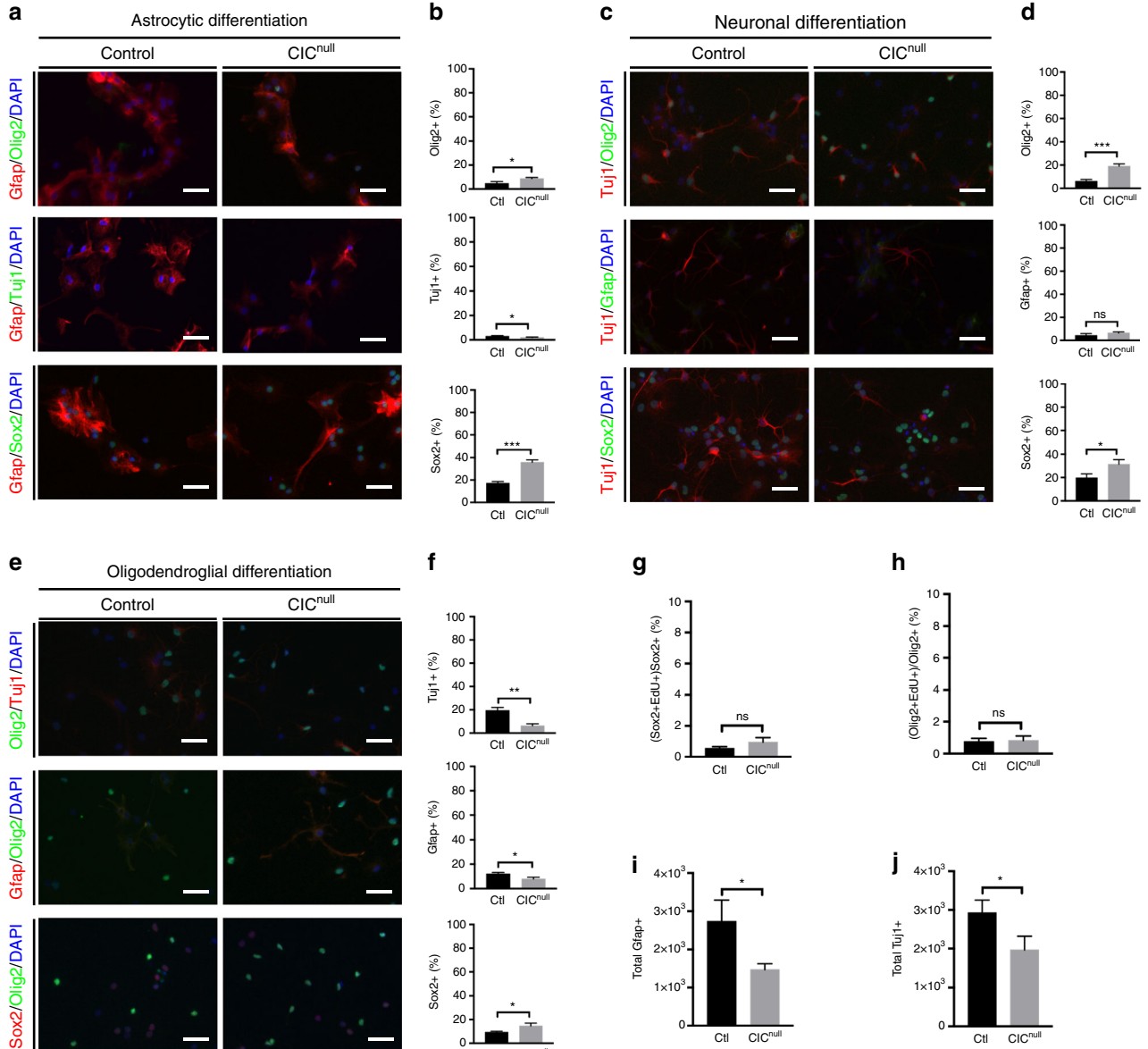

**Fig. 6** Alternate lineage selection of *Cic*-null NSCs in response to extrinsic differentiation cues. **a**, **b** NSCs cultured for 10 days in astrocyte-promoting media and stained for Sox2, Olig2, GFAP, and Tuj1. Representative images of cultures stained for oligodendrocytes (Olig2 + ), neurons (Tuj1 + ) and stem cells (Sox2 + ) in **a**; and corresponding quantifications of Olig2 + cells, Tuj1 + cells, and Sox2 + cells as a percentage of total cell enumerated by DAPI nuclear counterstain **b**. **c**, **d** NSCs cultured for 10 days under neuronal-promoting conditions, with representative images **c** of cultures stained for oligodendrocytes (Olig2 + ), astrocytes (Gfap + ) and stem cells (Sox2 + ), and corresponding quantifications **d** of Olig2 + cells, Gfap + cells, and Sox2 + cells as a percentage of total cells enumerated by DAPI. **e**, **f** NSCs cultured for 10 days under oligodendroglial-promoting conditions, with representative images **e** of cultures stained for neurons (Tuj1 + ), astrocytes (Gfap + ) and stem cells (Sox2 + ) and corresponding quantifications **d** of Tuj1 + cells, Gfap + cells, and Sox2 + cells as a percentage of total cells enumerated by DAPI. **g** Percentage of Sox2 + cells positive for EdU incorporation after 10 days exposure to neuronal-promoting conditions. **h** Percentage of Olig2 + positive for EdU incorporation after 10 days exposure to neuronal-promoting conditions. **i** Absolute counts of Gfap + cells/well after differentiating $4 \times 10^3$ NSCs in astrocytic conditions for 10 days. **j** Absolute counts of Tuj1 + cells/well after differentiation $4 \times 10^3$ NSCs in neuronal conditions for 10 days. Data from $n = 3$ biologic replicates per condition for all experiments. Source data are provided as a Source Data file. Statistical analysis performed by unpaired *t* test. Bars indicate mean ± SD. Scale bar: 10 μm. ns—not significant, *$p < 0.05$, ***$p < 0.001$

(Fig. 6c, d), with a small but non-significant increase also in the percentage of Gfap + cells in the *Cic*-null cultures (Fig. 6d). With respect to oligodendroglial-promoting conditions, Sox2 + cells were increased, whereas both Tuj1 + and Gfap + cells were reduced compared with controls (Fig. 6e, f). These data further support that *Cic*-deficient NSCs are less responsive to neuronal and astrocytic cues, and instead remain as stem cells or are oligodendroglial-biased.

Because we previously found that Cic loss increases NSC proliferation, it was important to exclude that the decreased neuronal and astrocytic fractions observed in our experiments were owing to a dilutional effect from greater NSC/OPC proliferation. In both conditions, after the 10-day differentiation protocol, only rare proliferating cells remained (Supplementary Fig. 4j). Of the Sox2 + and Olig2 + cells present in the neuronal or astrocytic conditions, < 1% were EdU +, and there were no significant differences between the proliferation of Sox2 + or Olig2 + cells in the *Cic*-null vs. control cultures (Fig. 6g, h). Moreover, when the data were analyzed for absolute numbers of cells rather than percentages of cells, we found fewer total Gfap + cells in the astrocytic conditions, and a fewer total Tuj1 + cells in the neuronal conditions in Cic-null cultures compared with controls (Fig. 6i, j). No differences in apoptosis were found between Cic-null and control cultures (Supplementary Fig. 4j). Together, these data indicate that the cell type skewing is not a result of increased NSC proliferation, but rather is owing to intrinsic fate bias.

Cumulatively, we conclude that Cic is a critical regulator of proliferation, self-renewal, and cell fate of NSCs. Loss of Cic expands the OPC pool by not only by increasing NSC proliferation but also by biasing their specification towards oligodendrocytes.

**Cic represses Ets factor expression in the forebrain.** As Cic is a transcriptional repressor, one mechanism for our findings is that Cic loss de-represses specific genes driving NSC proliferation and lineage selection. In this respect, *Etv* genes, encoding Ets-domain transcription factors, are candidate targets-of-interest. *Etv1,4,5* have been identified as Cic targets in various mammalian cells[12,31], and are overexpressed in ODGs (Supplementary Fig. 5f–h and[32]). Furthermore, previous studies have documented Cic occupancy at the *Etv5* promoter, at least in cerebellar tissue[33]. Consistent with a functional relationship between Cic and Ets expression in the forebrain, we found that NSCs and OPCs (the cells with the lowest levels of nuclear Cic), expressed the highest levels of Etv4 and 5 (Fig. 7f–k; Supplemental Fig. 5e). We also confirmed that in forebrain tissue, there is evidence of *Etv5* promoter occupancy by Cic (Fig. 7c).

If Ets factors are the mediators of the observed effects of *Cic* loss, we would expect their levels to be elevated in our experimental systems of *Cic* loss. Indeed, after 2 days of growth in oligodendroglial conditions, *Etv5* mRNA was increased ~ 37-fold, whereas *Etv4* transcripts were increased ~ 19-fold in Cic-null cells (Fig. 7a)—a finding also confirmed at the protein level (Fig. 7b). Etv1 levels, however were much lower, and did not significantly differ between *Cic*-null and -wildtype cells. Although both *Etv4* and *Etv5* were both de-repressed in the absence of Cic, because of the comparatively higher levels of Etv5 relative to Etv4, as well as previous studies implicating Etv5 in mediating glial fate decisions[34], we focused additional studies on this Ets factor. Electroporation of *Cic* shRNA into the telencephalic VZ resulted in upregulation of *Etv5* transcript in the electroporated patch within 48 h (Fig. 7d). Our *Cic*-floxed, Cre-electroporated brains also showed an increase in Etv5 protein in the electroporated patch (Fig. 7e). These data support that the PEA3 Ets transcription factors, particularly *Etv4* and *Etv5*, are transcriptional repressive targets of Cic in the forebrain.

**Etv5 de-repression mediates effects of Cic loss in NSCs.** To determine whether the proliferative increase and OPC bias that we observed with CIC ablation was mediated by Etv5, we first asked whether Etv5 overexpression alone was sufficient to mimic the phenotype of Cic deficiency. Electroporation of wildtype *Etv5* phenocopied the increased proliferation of CIC-null cells (Fig. 8g, h; Supplementary Figs. 5i, 6a). Similarly, *Etv5* overexpression acutely increased proliferation of cultured NSCs (Fig. 8k, Supplementary Fig. 6c). We also performed epistasis experiments introducing a dominant negative form (*DNETV5*) in which *Etv5* was fused to the *Engrailed* transcriptional repressor domain. Introduction of *DNETV5* or knockdown of *Etv5* with siRNA or shRNA reduced the proliferation of cultured *Cic*-deficient NSCs back to control levels (Fig. 8l; Supplementary Fig. 6d). In vivo, the increased proliferation observed upon deletion of *Cic* by cre electroporation was abrogated by co-electroporation with *DNETV5*, resulting in proliferation that was comparable to baseline (Fig. 8i, j compared with 8g, h; Supplementary Fig. 6b). Based on the extent of the effects of *Etv5* overexpression and *DNETV5* rescue in these assays, we conclude that the proliferative effects of Cic loss are largely driven by de-repression of Etv5.

With respect to the altered proportions of cell types that we had observed in *Cic*-deficient NSCs exposed to the different lineage-specific culture conditions, overexpression of wildtype Etv5 also phenocopied *Cic* loss in many respects. There were decreased fractions of Tuj1 + cells and Gfap + cells in the neuronal and astrocytic conditions, and increased fractions of Olig2 + and Olig2 + Pdgfra + cells in the oligodendrocytic condition when *Etv5* was overexpressed—although the severity of the changes was somewhat less marked with *Etv5* overexpression compared with Cic loss. *DNETV5* was also able to substantially (although incompletely) rescue the phenotype of *Cic*-deficient cells in these assays. In the neuronal and astrocytic conditions, *DNETV5* significantly increased the populations of Tuj1 + and Gfap + cells, respectively (Fig. 8a, b, d, e). The shifts in Olig2 + cells and Olig2 + Pdgfra + OPCs also returned to control levels (Fig. 8c, f). The MBP + fraction similarly showed a substantial although partial rescue (Fig. 8c, f). That the *DNETV5* rescue in several of the differentiation assays was significant but only partial suggests that other factors may contribute to the lineage specification effects. Nevertheless, the results point to *Etv5* as playing a large part in the lineage-biased phenotype.

Thus, Etv5 overexpression is both sufficient for inducing proliferation and cell fate bias effects, and is necessary for the proliferative dysregulation and lineage bias resulting from CIC loss.

**Etv5 blockade decreases tumorigenicity of human ODG.** The above studies provide a plausible framework for understanding how disruption of *Cic*, a cancer-associated gene, can perturb a cell's normal developmental trajectory. We then asked whether in established ODG, persistent loss of *CIC* or the resulting *ETV5* upregulation is required for maintaining malignancy. We used two patient-derived ODG cell lines, BT54 and BT88, to investigate this. Both lines harbor 1p19q co-deletion[35]. BT54 harbors a splice acceptor site mutation in the remaining *CIC* exon 6, whereas BT88 harbors a missense mutation in the remaining *CIC* exon 20[3]. In both BT54 and BT88, either stable re-expression of wildtype CIC or stable expression of *DNETV5* significantly reduced proliferation as measured by EdU incorporation (Fig. 9a, c, f). Sphere-forming ability was also decreased by *CIC* expression or by introduction of *DNETV5* (Fig. 9b, d, e). These anti-proliferative and anti-self-renewal effects were also seen using ETV5 shRNA (Supplementary Fig. 7a, b). Consistent with an ongoing requirement for *CIC* loss or ETV5 expression for

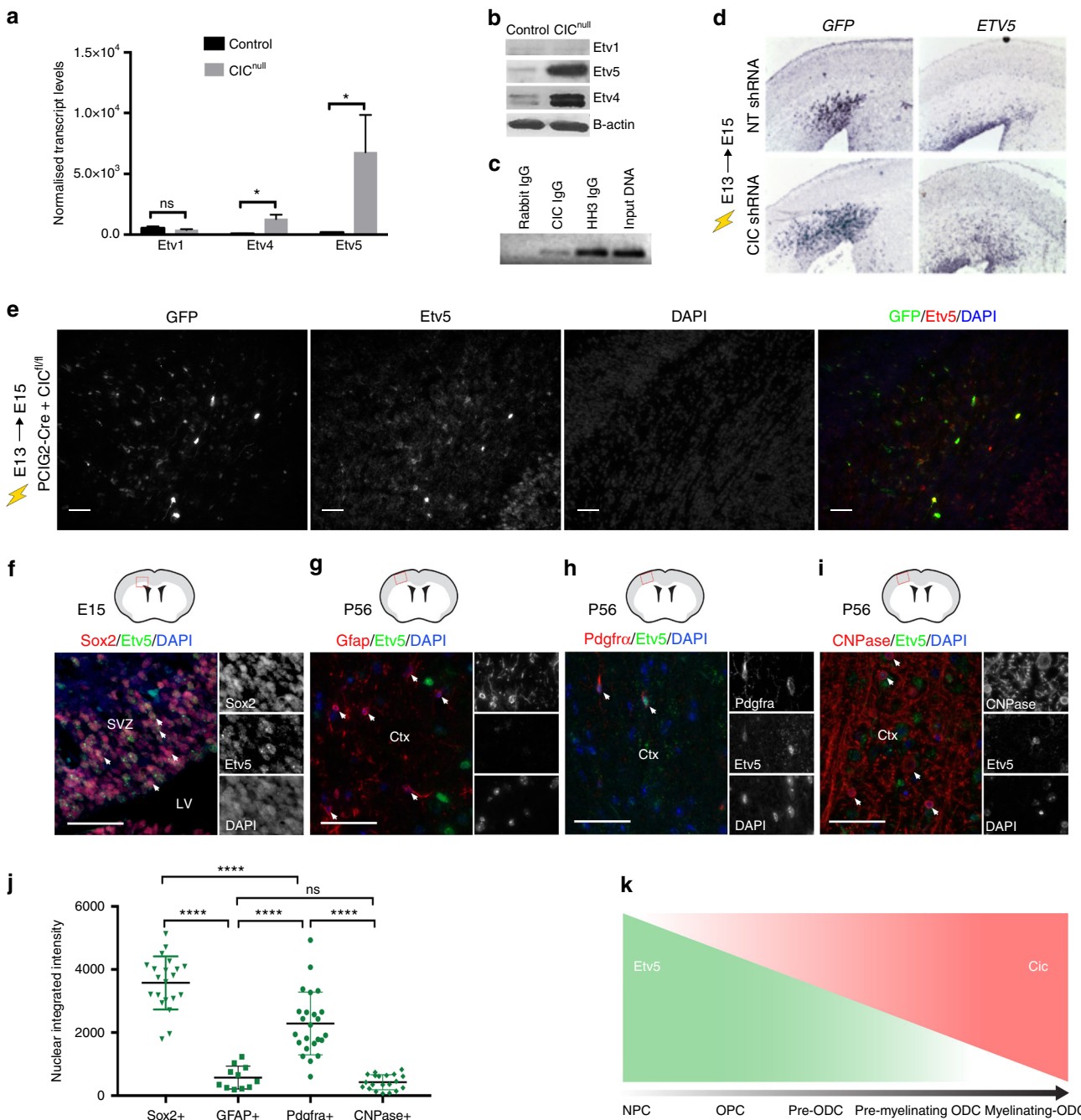

**Fig. 7** *Etv5* is a direct target of Cic transcriptional repression. **a**, **b** *Etv1*, *Etv4*, and *Etv5* mRNA expression levels **a** and protein expression levels **b** in cultured CIC-null and -control cells after 48-hour exposure to oligodendrocytic differentiating conditions; transcript levels normalized to average of three (Actin, GAPDH, Tubulin beta chain) housekeeping genes; data from $n = 3$ biologic replicates. Statistical analyses performed by unpaired $t$ test. Data shown as mean ± SD. *$p < 0.05$. **c** ChIP-PCR for Cic at the *Etv5* promoter. **d** Electroporation of Cic shRNA or non-targeting (NT) shRNA at E13 followed by in situ hybridization for *Etv5* and *GFP* at E15; *Etv5* transcripts are upregulated in areas of Cic knockdown indicated by the GFP expression; data from $n \geq 3$ mice per group. **e** Immunofluorescence staining for Etv5 and GFP protein expression 2 days after Cre electroporation into E13 VZ of *Cic*-floxed embryos. GFP + cells show increased Etv5 protein. Scale bar: 50 μm. **f** Immunofluorescence staining for Etv5 expression in Sox2 + NSCs at mid-neurogenesis. **g–i** Immunofluorescence staining for Etv5 expression Gfap + astrocytes **g**, Pdgfra + OPCs **h**, and CNPase + oligodendrocytes **i** in P56 adult cortex. Scale bar: 50 μm. **j** Quantitation of Etv5 nuclear staining for $n \geq 20$ cells for each cell type marker shown; each point represents one cell quantitated. Data from $n = 5$ mice per group. Data shown as mean ± SD. Source data are provided as a Source Data file. Statistical analyses performed with ANOVA with Tukey's post hoc test. ns–not significant, ****$p < 0.0001$. SVZ–subventricular zone, LV–lateral ventricle, Ctx–cortex. **k** Schematic depiction of relationship of Cic and Etv5 expression as neural stem cells differentiate to mature myelinating oligodendrocytes

proliferation and self-renewal in ODG, in vivo tumorigenicity of BT88 cells was markedly reduced, and survival of animals was increased, by stable expression of either wildtype *CIC* or *DNETV5* (Fig. 10). Thus, in cells that are already transformed to ODG, CIC loss and the subsequent elevation of ETV5 remains important to sustain the proliferative phenotype.

CIC loss and ongoing ETV5 elevation are also required for preventing oligodendroglial differentiation in established ODG.

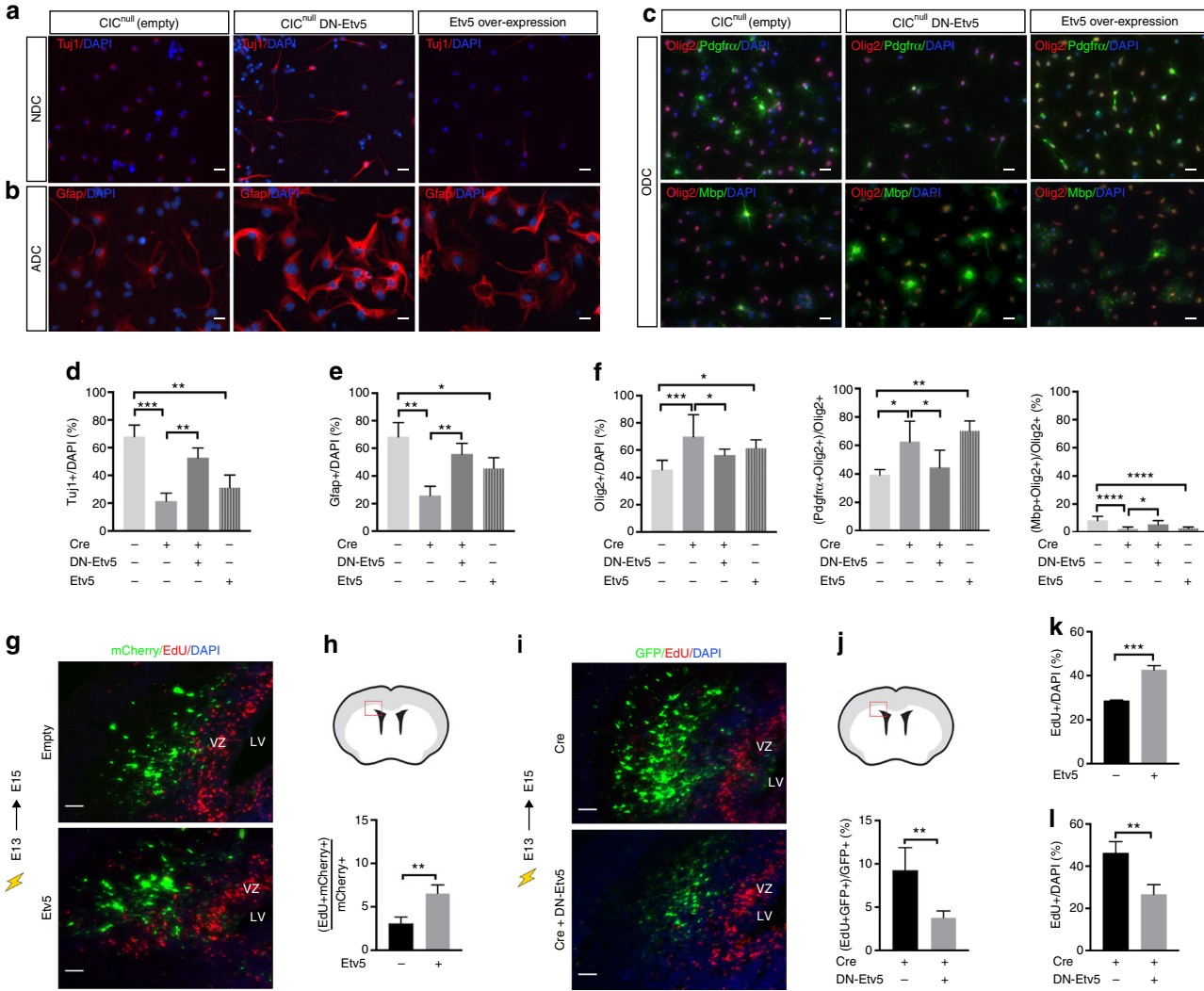

**Fig. 8** Etv5 is necessary and sufficient for proliferation and cell fate bias downstream of Cic loss. **a–f** Cic-null NSCs (Cic[null]Empty), Cic-null NSCs with dominant negative Etv5 (Cic[null]DN-Etv5), and Cic-wildtype NSCs overexpressing Etv5 (Etv5 overpression) were grown in lineage-directed culture conditions and assessed for their ability to differentiate to neurons, astrocytes, and oligodendrocytes as determined by immunostaining for bIII-Tubulin (Tuj1), Gfap, and Olig2, Pdgfra, and Mbp. Scale bar: 10 μm. Analysis of Tuj1 + cells from NSCs in neuronal differentiating condition, NDC **a**, **d**; analysis of Gfap + cells from NSCs in astrocytic differentiating condition, ADC **b**, **e**; and analyses of Olig2 + , Pdgfra + , and Mbp + cells from NSCs in oligodendrocyte differentiating condition, ODC **c**, **f** from n = 3 biological replicates, with three technical replicates each, for cell culture studies. **g**, **h** Representative images and quantitation of EdU incorporation 2 days post electroporation of wildtype *ETV5* or empty control plasmid, both carrying mCherry as a marker, into E13 CIC[Fl/Fl] VZ. Note: mCherry fluorescence and EdU staining were false-colorized to green and red after grayscale imaging. Scale bar: 50 μm. Data from n = 4 mice per each group. Scale bar: 50 μm. **i**, **j** Representative images and quantitation of EdU incorporation 2 days post-electroporation of *Cre* only or of *Cre* co-electroporated with DNETV5 into E13 CIC[Fl/Fl] VZ. Data from n = 4 mice per each group. Scale bar: 50 μm **k** EdU incorporation assay in cultured *Cic*-wildtype NSCs without or with ETV5 overexpression from n ≥ 3 biological replicates. **l** EdU incorporation in Cic-floxed NSCs with Cre, and without or with DNETV5 expression from n ≥ 3 biological replicates. Data shown as mean ± SD. Statistical analyses performed either *t* test in **h**, **j**, **k**, **l**; or with ANOVA with Tukey's post hoc test in **d**, **e**, **f**. ns–not significant, *p < 0.05, **p < 0.01, ***p < 0.0001. Source data are provided as a Source Data file. ADC–astrocytic differentiation condition, NDC–neuronal differentiation condition, ODC–oligodendrocytic differentiation condition. VZ–ventricular zone, LV–lateral ventricle

In response to oligodendrocyte differentiation conditions, ODG cells with *CIC* re-expression or *DNETV5* showed increased expression of CNPase compared with controls, indicating improvement in capacity for the tumor cells to differentiate and mature along the oligodendroglial progression (Figure 9g, h; Supplementary Fig. 7c, d). Interestingly, however, effects on neuronal and astrocytic differentiation were mixed. *CIC* re-expression increased the ability of cells to respond to neuronal differentiation cues by increasing expressing bIII-Tubulin (Tuj1 + ) and extending neurites; however this was not phenocopied by *DNETV5* (Fig. 9g, h; Supplementary Fig. 7c, d). With respect to expression of GFAP in response to astrocytic differentiation cues, no significant differences were detected with either *CIC* re-expression or *DNETV5* introduction compared with BT88 control (Fig. 9g, h; Supplementary Fig. 7c, d). The results suggest that although *CIC* loss and *ETV5* overexpression maintain ODG cell proliferation and self-renewal, and prevent progression of oligodendroglial differentiation in established ODG; other aspects such as the capacity for neuronal or astrocytic differentiation are determined by additional genetic (or epigenetic) alterations, at least in our culture system. The effects of CIC and ETV5 on ODG differentiation were not evaluated in vivo, however, as the lesions resulting from implantation of *CIC*- or *DNETV5*-expressing BT88 cells were too small to provide meaningful cell numbers for assessment.

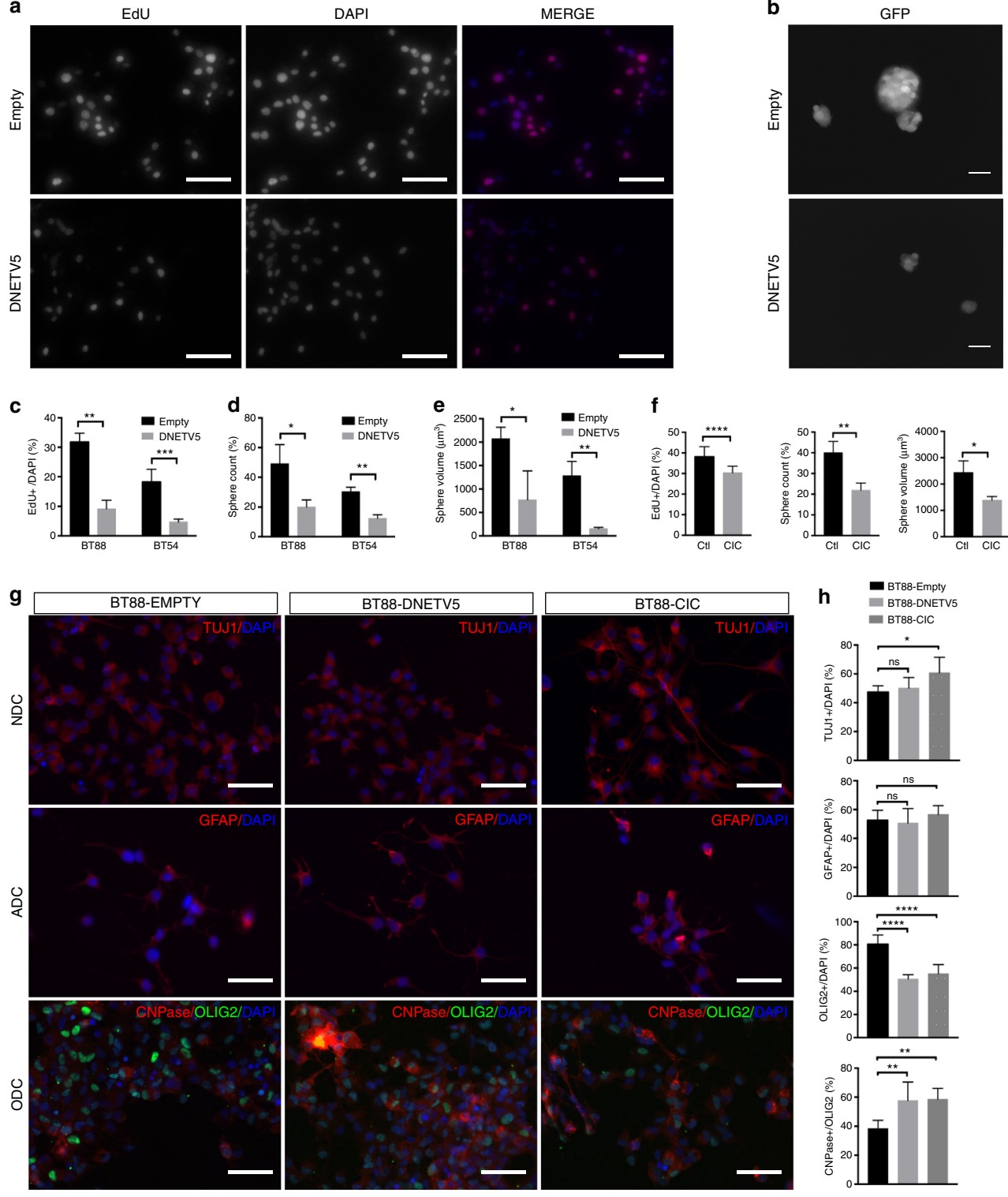

Overall, our results indicate that *Cic* regulates NSC proliferation and cell fate in neurodevelopment and ODG, and that the pro-proliferative and pro-OPC phenotypes observed with *Cic* loss are largely mediated through *Ets* transcriptional de-repression.

## Discussion

Genomic analyses of brain cancers have implicated *CIC* as a tumor suppressor gene in diffuse gliomas, particularly ODGs[3,4,36].

Here, we report for the first time cell type-specific differences in CIC expression in the brain. Moreover, our work provides new insight into the roles of this putative tumor suppressor in regulating the developmental fate of neural stem/progenitor cells.

We found that *Cic* loss biases neural stem cells away from selection of the neuronal lineage towards to selection of glial lineages. Adding to prior knowledge that RAS/MAPK pathway signaling and *ETV5* are both involved in gliogenic competence[17,34], our results firmly place *Cic* at the intersection of

**Fig. 9** ODG cells require CIC loss or elevated ETV5 to maintain proliferation and stemness. **a, c** EdU incorporation in BT88 and BT54 ODG cell lines stably transfected with control plasmid or with dominant-negative ETV5 (DNETV5). Representative images from BT88 cells **a** and quantitation of both BT88 and BT54 **c** from $n = 3$ biologic replicates each. Statistical analyses performed by $t$ test. **b, d, e** Neurosphere assays on BT88 and BT54 stably transfected with control plasmid or with DNETV5. Representative images showing effect of DNETV5 on sphere formation in BT88 ODG cells **b**, with quantitation of sphere number **d** and size **e** in both BT54 and BT88 ODG cells without and with DNETV5 after plating equal numbers of cells; data represent $n = 3$ biologic replicates. Statistical analyses performed by $t$ test. **f** CIC re-expression in BT88 cells decreases proliferation, sphere number, and sphere size; data represent $n = 3$ biologic replicates. Statistical analyses performed by $t$ test. **g, h** Lineage-specific differentiation capacity of BT88 ODG cells stably transfected with either control plasmid, DNETV5, or CIC from $n = 7$ biologic replicates. After 10-day exposure to neuronal differentiation condition (NDC), astrocyte differentiation condition (ADC), or oligodendrocyte differentiation condition (ODC), cells were analyzed for expression of Tuj1, GFAP, OLIG2, and CNPase by immunofluorescence **g**. Cell types were quantified as a percentage of total cell as indicated by DAPI nuclear counterstain **h**. Data from $n \geq 3$ biologic replicates. Statistical analyses by ANOVA with Tukey's post hoc test. Data shown as mean ± SD. Source data are provided as a Source Data file. All scale bars: 50 μm. ns–not significant, $*p < 0.05$, $**p < 0.01$, $****p < 0.0001$. NDC–neuronal differentiation condition, ADC–astrocytic differentiation condition, ODC–oligodendrocytic differentiation condition

two, providing a missing link between extrinsic differentiation signals and the execution of transcriptional programs critical for normal neuro- and gliogenesis. With respect to Cic and the anti-neuronal bias that we observed, recent work by Lu et al.[14] found that disrupting Ataxin1-Cic complexes resulted in abnormal maturation and maintenance of upper-layer cortical neurons—with effects on behavior, learning, and memory. Though our studies focused on early lineage selection events, and did not systematically address questions of cell maturation, some of our in vitro morphological findings are consistent with neuronal and oligodendrocyte maturation defects being part of the *Cic* phenotype. Future work would be needed to define the complex cellular dependencies on Cic, not only in the forebrain but also in the cerebellum where Cic pathology has also been implicated[37,38].

Our findings build on observations by Yang et al.[13] that *Cic* deficiency increases a population of proliferating OPCs in the brain. We now provide mechanistic foundations for those observations—identifying symmetric cell division alterations, lineage selection effects, and a downstream effector. These provide tangible links between *CIC* loss and ODG biology—as both dysregulated proliferation and the persistence of an immature OPC phenotype are cardinal features of this cancer. Many parallels exist between normal OPCs and the cells comprising ODG. In common, both OPCs and ODGs express PDGF, PDGFR, and NG2, which control OPC differentiation[39,40]. Together, the results may explain some of these ODG features, and are consistent with other reports, suggesting that dysregulation of OPCs or OPC-like cells are amongst the early changes in gliomagenesis[2,41]. There are some notable differences between the work by Yang et al.[13], however, and our studies. One difference is that the prior study used HOG cells that, despite their historic name, do not carry the ODG-defining genetic features of 1p19q loss or *CIC* mutation. Similarly, the *Pdgfra*-amplified/overexpressed mouse glioma model that was used is more akin to an RTK-driven GBM than ODG. In deploying the BT88 and BT54 models (which genetically and phenotypically recapitulate ODG), our work may be closer to clinical relevance for ODG. Another difference is that our studies also further clarify the early neuronal-glial fate decision effects of NSCs, with the bias away from neuronal fates and toward glial fates in the setting of CIC loss. Most importantly, however, we now identify *Ets* derepression as a key mechanism underlying the phenotypic effects of Cic loss in NSCs.

The observation that *DNETV5* could abrogate so much of the pro-proliferative and pro-OPC phenotype resulting from *Cic* loss was unexpected, and together with our expression and *ETV5* knockdown experiments, points to a critical role for this particular Ets factor. However, we do not exclude that other Ets factors such as Etv4 may contribute. The dominant-negative construct may not be entirely specific for Etv5; cross-reactivity has been

reported with similar approaches, and further experiments would be required to dissect the relative contribution of different Ets genes. Nevertheless, the identification of Ets factors as likely mediators is salient to the development of future therapies. Reports of Ets inhibition through peptidomimetic or small molecule approaches[42–44] lend hope that Etv5 could potentially be inhibited for clinical benefit for ODG.

Finally, though relevant to ODG, our studies do not directly model ODG. Our *Cic* conditional knockout mice, however, will be a useful tool for future cell type- and stage-specific *Cic* deletion in the context of other mutations such as of *IDH1*(R132H)[45]. We conceptualize *IDH1/2* mutations as a first event in ODG, causing epigenetic changes that expand the potential pool of cells vulnerable to transformation, with second-hit loss of *CIC* serving to dysregulate proliferation, bias cells to oligodendroglial lineage, and delay them in an immature state. Additional studies using mouse models and human ODG samples, however, are needed to further dissect the relationship between *CIC* and *IDH* mutations. Although one recent study used single-cell RNA-Seq to examine ODG cell subpopulations and did not detect any differences between ODG cells with and without *CIC* mutation aside from the increased expression of PEA3 Ets factors, the number of cells analyzed was limiting for resolution of cell type-specific differences, and functional studies had not been performed[46]. Similar approaches with larger cell numbers might provide further insight into the ODG biology with respect to *CIC*.

In summary, we show that Cic regulates proliferation, fate decisions, and differentiation in neural stem cells, with loss particularly expanding the OPC population. Furthermore, our findings indicate that disruption of the Cic-Etv axis is central to the biology of ODG.

## Methods

**Mice.** CD1 outbred mice (Jackson Labs) were used for Cic expression analyses. *Cic* conditional knockout (*Cic*-CKO) mice were generated at Taconic Artemis by homologous recombination to flank Cic exons 2–11 with loxP sites. The exons targeted are 2–11 of *Cic* short form, transcript variant 1, NM_027882.4 (equivalent to exons 3–12 of the *Cic* long form, NM_001302811.1, Cic transcript variant 4). The targeting vector was generated using clones from the C57BL/6 J RPCIB-731 BAC library, and consisted of a 4.0 kb 5′ flanking arm, neomycin resistance cassette (flanked by FRT sites), 5.5 kb loxP-flanked region, a puromycin resistance cassette (flanked by F3 sites), and 6.0 kb 3′ flanking arm. After homologous recombination in C57BL/6 N Tac ES cells, generation of chimeric animals, and germline transmission, NeoR and PuroR cassettes were removed via Flp recombination by breeding with a Flp deleter line. The final CIC-CKO allele carries loxP sites in introns 1 and 11, and single residual FRT and F3 sites. Expression of Cre recombinase results in deletion of exons 2–11 and frameshifting of the remaining CIC-S exons 12–20. Genotyping primers were as follows: CIC 5′ flanking region (192 bp wt, 348 bp CKO allele) 5′-AGGAGGTTGTTACTCGCTATGG-3′ (forward) and 5′-CTGATGTCCTAAGACCTTTACAAGG-3′ (reverse); CIC 3′ flanking region (273 bp wt, 410 bp CKO) 5′-CTTTGTCACTGTCTGCCTTCC-3′ (forward) and 5′-TGGGTAATACCACCGTGCC-3′ (reverse)'. FoxG1-cre mice (Jackson Labs) were bred to *Cic*-CKO mice to generate telencephalic CIC

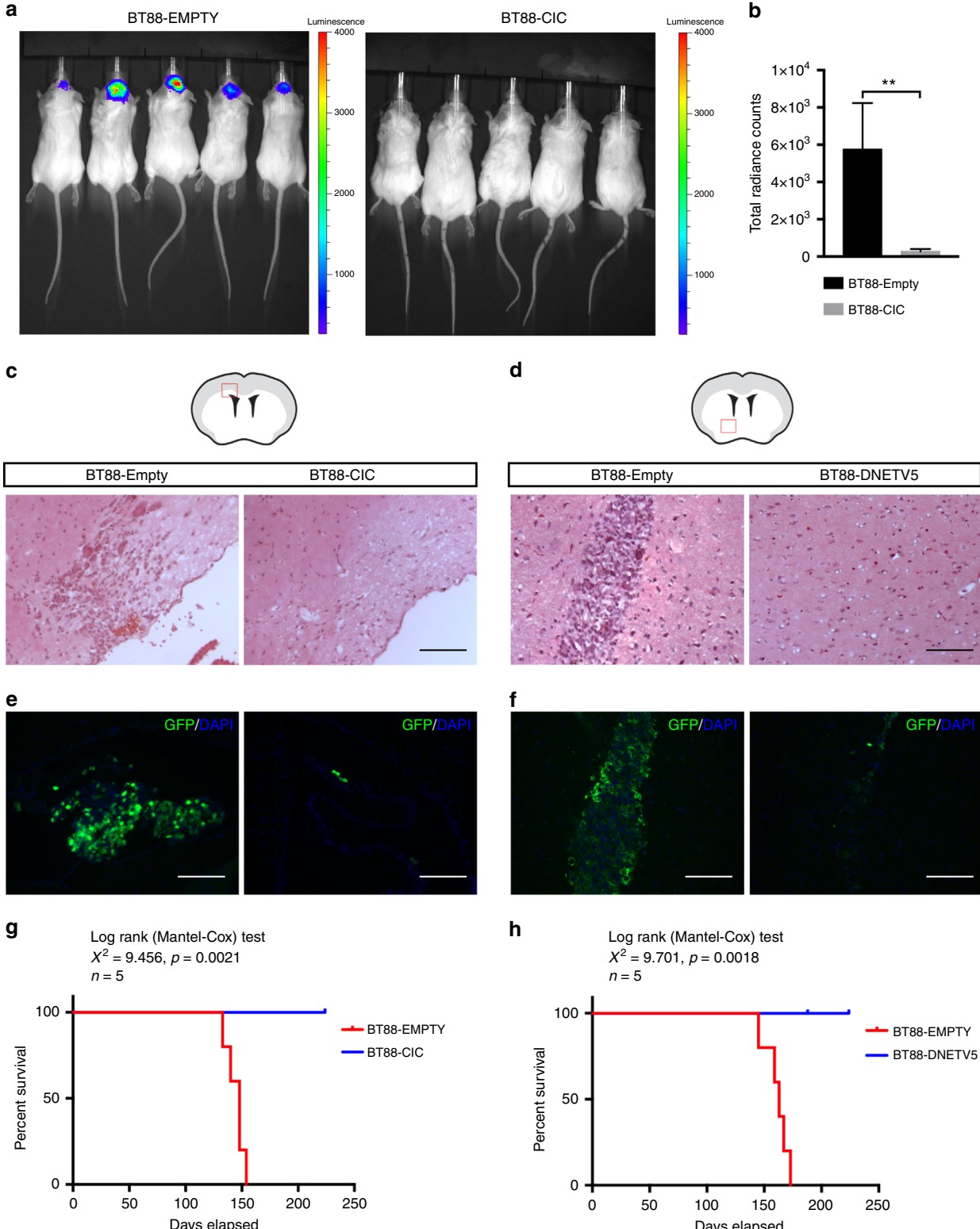

**Fig. 10** Tumorigenicity of ODG cells is reduced by CIC re-expression or ETV5 blockade. **a**, **b** Bioluminescence imaging of NOD-SCID mice at 6 weeks post-orthotopic implantation of BT88 oligodendroglioma cells stably transfected with empty-GFP-luciferase or CIC-GFP-luciferase. Representative images **a** and luminescence data **b** from five mice per cohort. Data shown as mean ± SD. Statistical analyses by student's $t$ test. **$p$ < 0.01. **c**, **e** Representative images of brain sections from mice implanted with control-GFP-luciferase BT88 cells or CIC-GFP-luciferase BT88 cells stained with H&E **c** or GFP **e** at 6 weeks post implant. Scale bars: 50 μm. **d**, **f** Representative images of brain sections from mice implanted with control-GFP-luciferase BT88 cells or DNETV5-GFP stained with H&E **d** or GFP **f** at 6 weeks post implant. Scale bars: 50 μm. **g** Kaplan–Meier survival analysis of mice implanted with BT88 cell transfected with empty-GFP-luciferase or CIC-GFP-luciferase. $n = 5$ mice per cohort. Statistical analysis by log-rank (Mantel–Cox) test; $p = 0.0021$. **h** Kaplan–Meier survival analysis for BT88 cell transfected with empty-GFP or DNETV5-GFP. $n = 5$ mice per cohort. Source data are provided as a Source Data file. Statistical analysis by log-rank (Mantel–Cox) test; $p = 0.0018$

knockouts and littermate controls. After killing, dissected brains were processed for histology, western blotting, or cell culture. Knockout in tissue or cells post-cre was confirmed by Q-RT–PCR using primer/probe sets for CIC (Applied Biosystems) and by Western blotting with anti-Cic. In all mouse experiments, the morning of vaginal plug was designated embryonic day 0.5 (E0.5). Both males and females were used.

**In utero electroporation.** In utero electroporation was performed as described previously using the following plasmids: pLKO.1-Cic shRNA (Sigma, TRCN0000304642; 5′-CCGGAGCGGGAGAAGGACCATATTCCTCGAGGAA-TATGGTCCTTCTCCCGCTTTTTTG-3′), pLKO.1-non-targeting shRNA (Sigma; 5′-CCTAAGGTTAAGTCGCCCTCGCTCGAGCGAGGGCGACTTAACCT-TAGG-3′), pCIG2-Cre (which contains Cre-IRES-GFP), pCIC-ETV5 (which contains Etv5-IRES-mCherry), Super piggyBac Transposase (Systems Biosciences, SBI), and piggyBac cargo vector PB513B-1 (SBI) into which cDNAs were cloned for Turbo-Cre and Etv5. The DNETV5 consists of an *Etv5-EnR* fusion (gift of Dr. Carol Schuurmans) cloned into the piggyBac construct modified to contain the CAG promoter and GFP-luciferase. DNA was prepared with Endo-free DNA kit (Qiagen) and was injected at 1.5 μg/μl into the telencephalic vesicles of embryos in time-staged pregnant females using a Femtojet 4i microinjector (Eppendorf) then followed by electrical pulses (6 × 43 V, 950 ms interval) applied by platinum tweezer-style electrodes (7 mm, Protech) uSsing a BTX square wave generator (Harvard Apparatus). Post-procedure, embryos were allowed to develop until the time of harvesting. EdU (50 mg/ml in phosphate-buffered saline; PBS) was injected intraperitoneally into the pregnant dam 30 min prior to killing.

**Neural stem cell culture.** The VZ of E15 brains were dissected, and tissue was dissociated to single cells using Accumax (EMD Millipore). Cells were grown at 37 °C, 5% $CO_2$ in low-adhesion tissue culture flasks (Sarstedt) in mouse neural stem cell (mNSC) media consisting of NeuroCult Proliferation media (Stem Cell Technologies) supplemented with heparin, epidermal growth factor (EGF 20 ng/mL; Peprotech), and fibroblast growth factor (FGF 20 ng/mL; Peprotech). When spheres reached 100–200 μm in diameter, cells were split using Accumax and re-plated at 20,000 cells/mL.

**Transfection of cultured cells.** In all, $1–4 \times 10^6$ dissociated cells were resuspended in 100 μL of Amaxa Mouse NSC Nucleofector Solution (VPG-1004, Lonza) with 5 μg of plasmid DNA. Nucleofection was performed with a Nucleofector II Device (Amaxa) using the A-033 program. Cells were returned to mouse neural stem cell media for further expansion/selection. For siRNA-mediated Etv5 knockdown, cells were transfected with 50 nM Etv5 or Etv4 ON-TARGET Plus SMARTpool siRNAs (Dharmacon) using Lipofectamine 3000 reagent per manufacturer's protocol. Cells were assayed 48 h hours post transfection.

**Trypan assay.** In total, 150,000 cells were seeded per T25 flask in mNSC media and counted after 72 h on a TC20 cell counter (Biorad) using Trypan blue stain (Thermo). Both dead and live cells were counted.

**Paired cell assay.** Dissociated cells were plated at 1000 cells/ml in mNSC media on dishes coated with CTS CELLstart (Thermo). After 20 h, cells were fixed in 4% paraformaldehyde (PFA) and immunostained for Ki67. Pairs were scored as symmetric proliferative if both daughter nuclei were Ki67 +, symmetric differentiative if both were Ki67−, and asymmetric if one nucleus was Ki67 + and the other Ki67−.

**Neural colony-forming cell assay.** Cells in semi-solid media were prepared using the Neurocult NCFC Assay Kit (Stem Cell Technologies) per manufacturer's protocol. Cells were plated at a density of 1650 cells/ml using 1.5 ml per 35 mm culture dish. Dishes were replenished after 7 days with 60 μl of neural stem cell proliferation media supplemented with heparin, EGF, and FGF. Sphere number and size were scored using a gridded scoring dish (Stem Cell Technologies). Eight-cell aggregates were used as the cutoff for scoring.

**Lineage-directed differentiation of NSCs.** To promote neuronal differentiation, NSCs were seeded in mNSC proliferation media (Stem Cell Technologies) on coverslips coated with Poly-L-Ornitihine and Laminin. After 24 h, media was replaced with Neurobasal Media, 2% B-27, 2 mM GlutaMAX-I (Thermo). After 4 more days dibutyryl cAMP (Sigma) was added daily to a final concentration of 0.5 mM. To promote oligodendroglial differentiation, NSCs were seeded on coverslips coated with Poly-L-Ornitihine and Laminin. After 24 h, media was replaced with Neurobasal media, 2% B-27, GlutaMAX-I, 30 ng/mL 3,3′,5-Tri-iodo-L-thyronine sodium (Sigma). To promote astrocyte differentiation, NSCs were seeded on coverslips coated with Geltrex (Thermo). After 24 h, media was replaced with DMEM, 1% $N_2$-Supplement, 2 mM GlutaMAX-I, 1% FBS. The cells were maintained in the respective differentiation media for 10 days prior to assays of lineage and differentiation.

**Immunostaining.** Tissue was fixed overnight in 4% PFA, cryoprotected in 30% sucrose/PBS, and embedded in Tissue-Tek O.C.T. (Sakura Finetek) prior to cutting 6 μm cryosections on Superfrost Plus slides (VWR). Cultured cells grown on coverslips were fixed for 20 min in 4% PFA then rinsed in PBS. Permeabilization was performed with 1 × TBST (Tris-buffered saline: 25 mM Tris, 0.14 M NaCl, 0.1% Triton X-100) for 15 min at room temperature (RT). Three percent% goat or horse serum in TBST × 30 min was used for block. Primary antibodies applied for 1 h at RT or O/N at 4 °C in block. Alexa Fluor secondary antibodies were applied at 1:500 dilution for 1 h RT. Nuclei were counterstained in 4′,6-diamidino-2-phenylindole (DAPI; Santa Cruz) and mounted with FluorSave Reagent (Calbiochem). For immunostaining of tissue sections when two or more primary antibodies were from the same host species and for comparison of cell type specific CIC expression, the Opal 4-color IHC kit (Perkin Elmer) was used per manufacturer's protocol.

**Western blotting.** Cells/tissue was lysed in RIPA buffer (10 mM Tris-Cl pH 8, 1 mM ethylenediaminetetraacetic acid, 1% Triton X-100, 0.1% sodium deoxycholate, 0.1% sodium dodecyl sulphate, 140 mM NaCl, 1 mM phenylmethylsulfonyl fluoride) with 1 × Halt Protease and Phosphatase Inhibitor Cocktail (Thermo). In total, 20 μg protein lysate (50 μg in the case of CIC probing) was run on 4–12% Bis-Tris or 3–8% Tris-Acetate gels (Thermo). Protein was transferred to polyvinylidene difluoride membranes in NuPAGE Transfer Buffer (Thermo). Membranes were blocked in TBST with 5% powdered milk. Primary antibodies diluted were applied for 1 h at room temperature or overnight at 4 °C. Horseradish peroxidase-coupled secondary antibodies were applied for 1 h at room temperature. Membranes were developed using ECL Plus Western Blotting Reagent (GE Healthcare) and X-ray film.

**Antibodies.** Primary antibodies directed against the following were used: CIC (Rabbit Polyclonal, 1:500 IF, 1:1000 WB; A301-204, Bethyl), CIC (Rabbit Polyclonal, 1:100 IF; PA1-46018; Thermo), Olig2 (Mouse Monoclonal, 1:250 IF, 1:1000 WB, MABN50, clone 211F1, Millipore), Olig2 (Rabbit Polyclonal, 1:500, AB9610, Millipore), ETV1 (Mouse Monoclonal, 1:1000 WB, SAB1403794, clone 4C12, Sigma), ETV4 (Rabbit Polyclonal, 1:2000 WB, LS-C98380, LS BioSciences), ETV4 (Rabbit Polyclonal, 1:500 IF 1:2000 WB, ARP32263_P050, AVIVA Systems Biology), ETV5 (Rabbit Polyclonal, 1:1000 WB; sc-22807, Santa Cruz), ETV5 (Polyclonal Rabbit, 1:500 IF, 1:1000 WB, 13011-1-AP, ProteinTech), GFP (Rabbit Polyclonal, 1:500 IF; A11122, Thermo), GFP (Mouse Monoclonal, 1:250 AF, ab38689, clone 6AT316, Abcam), SOX2 (Rabbit Monoclonal, 1:500 IF, ab92494, clone EPR3131, Abcam), Turbo-GFP (Mouse Monoclonal, 1:500 IF, TA150041, clone OTI2H8, Origene), SOX2 (Rabbit Polyclonal, 1:500 IF, 1:1000 WB, #3728, clone C70B1, Cell Signalling), SOX9 (1:500 IF, ab76997, clone 3C10, Abcam), SOX9 (Rabbit Polyclonal, 1:250 IF,1:1000 WB, AB5535, Millipore), PDGFRA (Rabbit Monoclonal, 1:500 IF, 1:1000 WB, 3174 S(D1E1E), Cell Signalling), PDGFRA (Goat, Polyclonal, 1:250 IF, AF1062, R&D), CC1 (1:500 IF, APC (Ab-7) (OP80, clone CC1, Millipore), MBP (1:500 IF, ab40390, Abcam), Nestin (1:500 IF, 1:1000 WB, MAB353, Millipore), Tbr1 (1:500 IF, AB2261, Millipore), Tbr2 (Mouse Monoclonal, 1:500 IF, AB15894, Millipore), Neuron-specific beta-III Tubulin (Mouse Monoclonal, 1:500 IF, 1:1000 WB, MAB1195, clone Tuj1, R&D), GFAP (Mouse Monoclonal, 1:800 IF, 1:1000 WB, MAB360, clone GA5, Millipore), GFAP (Rat Monoclonal, 1:500 IF, 345860, clone 2.2B10, Millipore), ALDH1 (Rabbit Polyclonal, 1:250 IF, ab87117, Abcam), NeuN (Mouse Monoclonal, 1:500 IF, MAB377, clone A60, Millipore), Cre (Rabbit Polyclonal, 1:500 IF, 1:2000 WB, NB100-5613, Novus), Ki67 (Rabbit Polyclonal, 1:200; ab15580, Abcam), EdU (61135-33-9, Carbosynth US LLC). For routine immunofluorescence, secondary antibodies were: Alexa Fluor-488, -594 and -633) conjugated species-specific antibodies (Thermo) were used at 1:500 dilution. For the OPAL method (OPAL 4-color kit, Perkin Elmer), secondary antibodies used were anti-mouse HRP polymer (DAKO) and anti-rabbit HRP polymer (DAKO).

**Transcriptional analysis.** Total RNA was extracted using AllPrep DNA/RNA/Protein Mini Kit (Qiagen) per manufacturer's protocol. For each of three biologic replicates per condition, 100 ng RNA was subjected to nanoString analysis on nCounter system (NanoString Technologies) using a custom gene expression codeset containing neurodevelopmental and brain cancer associated genes, and three housekeeping genes. Data were analyzed using nSolver software and exported to PRISM software for further statistical analyses. Counts for Etv1, Etv4, and Etv5 were normalized to the average of three housekeeping genes (GAPDH, Actin, Tubulin beta chain).

**In situ hybridization (ISH).** Digoxygenin (DIG)-labeled riboprobes were prepared using a 10 × DIG-labeling kit (Roche), and ISH was performed as described previously using probes for *GFP* and *Etv5*[17].

**Chromatin immunoprecipitation.** In total, 50 mg P6 mouse forebrain tissue was use per chromatin immunoprecipitation (ChIP) assay. ChIP was performed with the SimpleChip Plus Kit with magnetic beads (Cell Signalling) per manufacturer's protocol with the following exceptions. Tissue was disaggregated using a Dounce homogenizer. In place of micrococcal nuclease, shearing was performed on a UCD Bioruptor on high power with intervals of 30 s on, 1 min off for 30 min. A total of

1.5 μl of anti-CIC (PA1-46018, Thermo) was used per immunoprecipitation reaction. Histone H3 antibody and normal rabbit IgG were used as positive and negative controls, as supplied in the kit. PCR primers for the ETV5 promoter region were: 5′-GGTGCAGGCCGAGGCCAGGG-3′ (For) and 5′-CATTGACCAATCAGCACCGG-3′ (Rev).

**Image analysis**. Coronal sections at the level of the anterior commissure were used in all quantitations of cell populations and staining intensities in the dorsal cerebral cortex and corpus callosum. Slides were imaged on AxioObserver fluorescence microscope (Zeiss). Images were processed using CS6 Photoshop software (Adobe) for orientation, false colorization, and overlay/colocalization. Enumeration of cells positive for cytoskeletal, cytoplasmic or membrane proteins was performed manually by counting positive cells using the Photoshop CS6 counting tool. Quantitation of CIC nuclear staining intensity and MBP staining density was performed using FIJI software[47] as follows. Images were first processed using Gaussian blur, then background subtracted. Default threshold limits were used in the threshold tool. Images were converted to binary and watershed was run to separate clumped cells. Nuclear staining was quantitated by using the analyzing particles option with separate cutoffs set for each antibody used. For quantitation of MBP expression on tissue sections, FIJI was used by drawing regions of interest on the lateral corpus callosum (cingulum), and the mean integrated density in the regions of interest was calculated.

**Intracranial xenografts and bioluminescence imaging**. BT88 ODG cells[35] were obtained from the Dr. S. Weiss, University of Calgary. A total of $1 \times 10^5$ BT88 cells stably transfected with CIC-GFP-Luciferase or DNETV5-GFP or respective empty vectors were stereotactically implanted into the right striata of 6- to 8-week-old NOD/SCID mice. Six weeks post implantation, tumor burden was measured either by bioluminescence imaging using IVIS Spectrum In Vivo Imaging System (Xenogen) and/or by histology from killed animals. Mice were anaesthetized under isoflurane and intraperitoneal injection of XenoLight D-Luciferin (Perkin Elmer) was administered at a dose of 150 mg/kg body weight. Acquisition of bioluminescence images was performed 10 min post injection. Analysis was performed using Living Image Software by measurement of photon counts (photon/s/cm²) with a region of interest drawn around the bioluminescence signal.

**Statistics**. Data are represented as mean ± SD from at least three biologic replicates for experiments. Comparisons between experimental and control samples were made using two-tailed $t$ test or, when there were more than two groups, using ANOVA. Tukey's procedure was applied post hoc to correct for multiple comparisons during multiple pairwise analyses (e.g., differential Cic expression among cell types). Bonferroni post hoc correction was applied for statistical analysis of data from NanoString assays. Survival curves were analyzed by log-rank (Mantel–Cox) test. Statistical analyses were performed using Prism software (Graphpad).

**Study approval**. Animal use was approved by the University of Calgary Animal Care Committee (protocol AC16-0266) in compliance with the Guidelines of the Canadian Council of Animal Care. Collection and generation of the glioma cell lines was approved by the Health Research Ethics Board of Alberta (protocols HREBA.CC-16-0762 and HREBBA.CC-16-0154).

**Reporting summary**. Further information on research design is available in the Nature Research Reporting Summary linked to this article.

## Data availability
The authors declare that all data supporting the findings of this study are available within the article and its Supplementary Information files. All biological materials used in this study are available from the corresponding author upon reasonable request.

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

## Acknowledgements

This work was funded by Canadian Institutes for Health Research (J.A.C.), Cancer Research Society (C.S., J.A.C.), Alberta Cancer Foundation (S.T.A., A.D.R., L.F., J.G.C.), Alberta Innovates Health Solutions (J.A.C., C.S.). We thank Edwin Herrenschmidt of Spine Visual Design for original graphic artwork.

## Author contributions

Conceptualization, J.A.C., S.T.A.; methodology, J.A.C., C.S., S.T.A.; investigation, S.T.A., A.D.R., R.D., L.F., M.J.C., M.A., W.W., S.O.L., M.D.B.; acquisition, analyses, and interpretation of data, J.A.C., S.T.A., A.D.R.; writing—original draft, J.A.C., S.T.A.; writing—review & editing, J.A.C., C.S., J.G.C., S.M.R., M.A.M., W.W., R.D., S.O.L.; funding acquisition, J.A.C., J.G.C., C.S.; resources, L.A., M.A., C.S; supervision, J.A.C., C.S., J.G.C., S.M.R.

## Additional information

**Competing interests:** The authors declare no competing interests.

