## [Peer Review File · Nature Communications]

Reviewers' comments:

Reviewer #1 (Remarks to the Author):

In this article by Chan and colleagues the role of *Capicua* is described in the developing brain. Using a combination of mouse genetics and in vitro stem cells systems, they show that loss of *Cic* promotes cell proliferation and pushes cells towards an OPC lineage. Mechanistically, they convincingly show that *Cic* directly represses *Etv5* expression and that some of the proliferative effects of *Cic*-loss are mediated by *Etv5*.

Generally, this is a very nice paper that describes, for the first time, the role of *Cic* in the developing brain. Below are a set of comments aimed at improving and clarifying the central messages of this paper:

1) Given that *Cic* seems to repress stem cell and OPC proliferation, it would be good to assess its expression in these populations in the adult. How generalizable is the low *Cic*-expression phenomenon to these other stages and systems? Is non-nuclear *Cic* found in all neural stem cells?

2) In figure 3 the IUE of *Cre* into the *Cic*-Floxed mouse demonstrates increased proliferation 2 days after IUE. This is a striking effect on proliferation, which raises the question of what happens to neurogenesis? If the cells are stuck in a rapidly dividing state, I would imagine that neurogenesis will also be dysregulated. If so, then how specific is the *Cic*-null phenomenon to OPC biology? Could it just be that loss of *Cic*-promotes stem cell proliferation and that OPC phenomenon is secondary to this effect on stem cells? (See my comment on *Sox9*).

3) The interpretation that *Sox9* is evidence of an oligo-bias is incorrect. *Sox9* has been linked to promotion of neural stem cell formation (Scott, et al Nature Neuroscience, 2010) and is expressed exclusively in astrocytes (Sun, et al. Journal of Neuroscience, 2017). Given the defined and established role for *Sox9* in astrocyte fate (Kang, et al. 2012, Stolt, et al. 2003, and Taylor, et al. 2007) and its role in NSC (and a lot of other stem cell systems), I think the correct interpretation of the NSC experiments is that loss of *Cic* likely promotes NSC proliferation and not OPC's specifically. The increase in OPC production is likely due to the fact that there are more NSCs.

4) The conditional *FoxG1-Cre* experiment also demonstrates a clear increase in OPCs, however it is impossible to decipher whether these effects are specific to OPCs or are secondary to increases in neural stem cells. They should do a comprehensive timecourse in the *FoxG1-Cre* line and assess NSC proliferation, so that this figure shows both OPC and NSC data from the in vivo system. Also, if they want to claim an OPC effect, they should use an OPC-specific *Cre*, like *Sox10-Cre* or *Olig2-Cre*. Finally, they should also assess neurogenesis and astrocyte production in their existing lines and determine whether the changes they observed with these lineages in vitro are also observed in vivo, across a comprehensive timecourse.

5) The link to *Etv5* is convincing---it's very clear that *Cic* directly represses *Etv5*. However, *Etv5* function has been linked to astrocytes, not oligodendrocytes (see Fig. 5M of Li, et al. Neuron, 2012). This is a fundamental problem that is not at all consistent with their model.

6) The epistasis test should be performed with a more robust and specific method of inhibition (like shRNAi) as the DN-ETV is likely to inhibit all ETS-family members. Along the same lines, when they overexpression ETV5 in fig5L, it has a nice effect on proliferation. What happens to cell fates? Does overexpression of ETV5 make OPCs? Astrocytes? NSCs? This set of experiments is very important and cellular outcomes should be fully analyzed.

Reviewer #2 (Remarks to the Author):

In this study, Ahmad et al. investigate the developmental role of the transcriptional repressor Cic in neural stem cells and in neural progenitors. In particular, the authors investigate the role of Cic in cell type specification and in cell fate regulation. The main findings demonstrate that Cic is expressed in both astrocyte and neuronal nuclei, but at lower levels in stem cells and in oligodendrocyte (OL) lineage cells. The authors generated a Cic floxed allele, and induced localized Cic deletion by electroporation of a Cre expression vector. Targeted Cic deletion in the forebrain increased proliferation and self-renewal of neural stem/progenitor cells. In vitro analysis of Cic-null NSCs demonstrated that Cic loss biases cells to the OL lineage, but maintains cells in an OPC state and delays their maturation. Finally, the authors demonstrate that these effects depend on de-repression of Etv5.

This is a potentially interesting study, revealing a new and early developmental role for a gene involved in oligodendroglioma formation. The information provides an appealing explanation for the oligodendroglial lineage bias arising from Cic loss in neural stem cells, beginning with loss of asymmetric division. However, there are many significant issues, the main one being that most of the study was performed in cultured cells and not in vivo. This significantly limits the impact and significance of the findings. Although the in vitro results are consistent with a cell specification role of Cic in neural cells, there are too many unanswered questions both in the in vivo and the in vitro part of the study, which make it largely incomplete – and unfocused. Neither the cell specification aspect, nor the role of Cic in OPC development (proliferation and differentiation) are investigated in detail. Finally, the role of Etv5 was not investigated in vivo.

Main issues

1. Cic expression in NSCs. This should be investigated more thoroughly both in NSCs and in their progenies in the SVZ, in the rostro-migratory stream (neuroblasts and NSCs) and in white matter OPCs derived from SVZ NSCs. More SVZ cell markers should be used, e.g. Nestin, Olig2, etc. Cell quantification (% of Cic+ cells expressing different markers) should also be obtained. Characterization shown in Figure 1 is superficial and not quantitative.
2. The authors claim that SVZ analysis was performed in “adult” brain. However, the brains were analyzed at P21, which is not even young adult. Analysis should be performed at P90. How is Cic expression regulated in the adult SVZ as compared to the early postnatal SVZ (P7)? Do the levels change? Does Cic expression change in different progenitor cell types with age?
3. Figure 4. Analysis of gliogenesis. The characterization of the proliferation abnormalities induced by loss of Cic is rather complete, and it involves both in vitro and in vivo analysis. However, the effects on cell specification were exclusively investigated in cultured cells. This is a big gap in the study, as the effects on cell differentiation are truly novel.
4. The authors show that Cic null cells display different lineage specification from control cells, in particular they express higher levels of Sox9 and Olig2 – i.e. Cic loss induces a pro-oligodendroglial program. This is only analyzed in culture and not in vivo. Differentiation defects were investigated by using typical OL lineage markers that identify different developmental stages. How does Cic ablation affect Sox2+ cells, since Sox2 has been reported to regulate the oligodendroglial lineage?
5. Are Dcx neuroblasts not formed or apoptotic, and are differentiation abnormalities found in Cic-

deficient cells in vivo? Do OPC differentiation abnormalities impact myelination? Given the lethality of Foxg1-targeted Cic ablation, an OPC-specific Cre driver is needed.

6. It should be determined whether manipulation of Etv5 function in vivo can rescue the phenotype of the Cic-null mutant. The authors show that: i) Etv5 and 4 are upregulated after Cic knockdown, ii) DN-Etv5 lowers OPC and Olig2 cells, partially relieves differentiation block and reduces overall numbers of cells in Cic null cells, and iii) Etv5 overexpression causes NSC proliferation in culture. Would DN-Etv5 affect fate decisions in Cic-deficient NSCs? A link between these events should be established by demonstrating that knockdown of Etv5 in Cic-null mutant cells can rescue the phenotype in vivo.

Reviewer #3 (Remarks to the Author):

Ahmad et al. show in their manuscript "Capicua regulates neural progenitor cell proliferation and oligodendrocyte differentiation" an involvement of the Cic protein in the proliferation and differentiation properties of oligodendrocyte progenitor cells (OPCs) also implicating a possible role of Cic and OPCs in oligodendroglioma. By a newly generated Cic-flox mouse line, they could show in vivo and in vitro that deletion of Cic leads to increase in proliferation and in number not only of neural stem cells but also of OPCs through a switch of the division mode. Embryonic in vivo deletion lead to the development of a smaller forebrain as well as an increase in number of OPCs that they were stalled in their differentiation. At the end, the authors recognized EtV genes as important transcription factors mediating the OPCs differentiation process. Although interesting, the manuscript is in its current version pretty immature for publication in Nature Communications.

Major comments:

1. The authors claim that embryonically Cic is expressed in Sox2+ cells, however it is not only restricted to these cells. What other cells express cic? In the same line, at P21 Cic is low expressed in neural stem cells and high in differentiated cells. Are other progenitor cells in other areas, e.g. OPCs in the cortex and stem cells/transit amplifying progenitors in the dentate gyrus expressing Cic? To further support their hypothesis it would be nice to see that proliferating OPCs in the cortex do not express Cic in contrast to the non-proliferating OPCs that should express it.

2. At P21, the authors use GFAP as a marker for astrocytes in the grey matter of the cerebral cortex. However, it is known that only reactive astrocytes in the grey matter (except in layer I) express GFAP- It would be nice to see another more ubiquitous astrocytic marker, e.g. S100b, Glast, Aldh1 etc. Similarly, using Olig2 as an oligodendrocyte lineage marker is ok, but for the purposes of the manuscript it would be nice to separate between mature oligodendrocytes and OPCs by using specific markers, e.g. NG2, PDGFRa for OPCs, CC1, Gstpi, CNPase for mature and MBP, MAG, PLP for myelinating oligodendrocytes. This would be of great importance as the Olig2-staining was not very successful (unfortunately there is only one Olig2+ cell shown). An additional staining in areas with high numbers of cells of the oligodendrocyte lineage would be very useful.

3. The experiments with Cic deletion have to be described and analyzed more in detail. For example, in the in utero electroporated animals and in the ex-vivo transfected cells: -What is the identity of cells that were electroporated/transfected? What was the transfection rate? What is the progeny of the electroporated cells (longer survival times would be nice)? Do only the electroporated/transfected cells (of which identity?) proliferate more or also the neighboring non-electroporated cells? This would be of importance to identify any cell non-autonomous effects. In the

figure 3A it is very hard to see anything; higher power images would be extremely helpful.

4. In the in vitro experiments with the NSC-media, when the increase in Olig2-positive cells could be observed, it would be important to separate between OPCs, mature and myelinating oligodendrocytes. To further highlight the implication, do the authors also observe higher numbers of Olig2-positive cells in the neurospheres?

5. The results of the cell cultures with the different media are very interesting and the decrease in neuronal and astrocytic markers under the respective culture conditions nice. However, it would be nice to know what the identity of the other cells is. Do the cells become oligodendrocytes or do they remain NSCs? In the same line, further markers for astrocytes are needed as GFAP is a marker for reactive astrocytes. It could namely be that the observed effects are the result of (lack of) activation in these cells. Further, are the numbers of cells and their proliferation also changing? One would expect that subsets of NSC and/or OPCs (would be nice to know which one) are proliferating more while the ones primed to differentiate still do so. That would lead to a decrease in the proportion of neurons/astrocytes although the numbers remain constant.

Another important issue would be how far the cells of the oligodendrocyte lineage are differentiating in the B27/TIT conditions. Do the cells remain in the progenitor state or do they differentiate?

6. What is the phenotype of the mutant mice in regard to astrocytes, neurons and oligodendrocyte numbers? This would be important to understand if all the changes in the forebrain are due to proliferation changes in the OPCs or as the result of fate switch. What is the reason of the lethality?

7. In figure 5L they do not show any effects of Etv5 on proliferation as they claim in the text (line 315). The effects in cell numbers could also be due to survival differences and not due to changes in proliferation.

Minor comments

1. The numbers on line 275/276 do not correlate with the numbers in Fig, 4K. Similar on line 276/277 with fig. 4L. I assume that in the one case is proportion of all cells, in the other of Olig2-positive cells. Please clarify in the text.

2. Please clarify what cell culture conditions were used for the ETV4/5 experiments.

3. Do oligodendroglioma also express higher levels of Etv5 as the results would implicate?

Point-by-point response to reviewer comments:

Reviewer #1

In this article by Chan and colleagues the role of Capicua is described in the developing brain. Using a combination of mouse genetics and in vitro stem cells systems, they show that loss of Cic promotes cell proliferation and pushes cells towards an OPC lineage. Mechanistically, they convincingly show that Cic directly represses Etv5 expression and that some of the proliferative effects of Cic-loss are mediated by Etv5. Generally, this is a very nice paper that describes, for the first time, the role of Cic in the developing brain. Below are a set of comments aimed at improving and clarifying the central messages of this paper:

1) Given that Cic seems to repress stem cell and OPC proliferation, it would be good to assess its expression in these populations in the adult. How generalizable is the low Cic-expression phenomenon to these other stages and systems? Is non-nuclear Cic found in all neural stem cells?

Response: We have now examined CIC expression in the adult brain. We have also included additional markers for cell stages of Olig2+ cells to show expression specific to populations that are also positive for Pdgfra, CNPase, and CC1. Likewise, during the neurogenic period, we show Tbr2, Tbr1, and NeuN. Astrocytes are now analyzed by both GFAP and Aldh1. These data showing more complete characterization of CIC expression are now found in a revised Figure 1 and in a new Figure S1.

With respect to non-nuclear CIC (which, for technical reasons is more difficult to quantitate), we observe that all stem cells have some detectable level of non-nuclear CIC but that NSCs at embryonic ages appear to have more intense cytoplasmic CIC than NSCs at later ages. We do not make a large point of this, as the focus of our work is on the transcriptional repressor role.

2) In figure 3 the IUE of Cre into the Cic-Floxed mouse demonstrates increased proliferation 2 days after IUE. This is a striking effect on proliferation, which raises the question of what happens to neurogenesis? If the cells are stuck in a rapidly dividing state, I would imagine that neurogenesis will also be dysregulated. If so, then how specific is the Cic-null phenomenon to OPC biology? Could it just be that loss of Cic-promotes stem cell proliferation and that OPC phenomenon is secondary to this effect on stem cells? (See my comment on Sox9).

Response: Indeed we do think that the CIC-null phenomenon is not exclusive to OPC biology and that a major effect is at the level of the stem cell. We apologize that we did not make the two possibilities (which we believe are both possible and not mutually exclusive) as distinct as they should have been described. Our previously submitted manuscript was meant to draw attention to the stem cell stage, but in our original submission (which was co-submitted with another manuscript that focused on OPC proliferation), we had thought that including some data and discussion around the OPC phenomenon was a way to tie the two complementary co-submitted manuscripts together. Based on the comments received back from both reviewers 1 and 2, however, we realize that including this approach had diffused much of our manuscript's intended central message that we find CIC important in stem cell proliferation and cell fate specification. We have made changes throughout to better clarify our interpretation. In the revised manuscript, we do not focus on the OPC phenomenon, and we provide more

of our data on the NSC proliferation and lineage to provide more focus. One of the changes in the revised manuscript that speak to more solid focus on the NSC effects is that we show a more even treatment of specification of the three major lineages (not only our data on the OPC specification).

To answer the reviewer's question regarding neurogenesis, we do see alterations in neurogenesis. We provide in vivo data showing a decrease in Tbr1+ cells generated in our IUE experiments (which we show in Fig 4). This is also supported by the small but significant decrease in NeuN+ cells observed in our CIC floxed--FoxG1cre mice (now quantitated in Fig2), as well as our in vitro experiments showing the outcomes of cells under directed differentiation conditions (Figs 5,6 and Suppl Fig S3).

3) The interpretation that Sox9 is evidence of an oligo--bias is incorrect. Sox9 has been linked to promotion of neural stem cell formation (Scott, et al Nature Neuroscience, 2010) and is expressed exclusively in astrocytes (Sun, et al. Journal of Neuroscience, 2017). Given the defined and established role for Sox9 in astrocyte fate (Kang, et al. 2012, Stolt, et al. 2003, and Taylor, et al. 2007) and its role in NSC (and a lot of other stem cell systems), I think the correct interpretation of the NSC experiments is that loss of Cic likely promotes NSC proliferation and not OPC's specifically. The increase in OPC production is likely due to the fact that there are more NSCs.

Response: We agree that Sox9 is not oligo specific, but submit that Sox 9 is not limited to astrocytic roles. While we are aware of the papers defining its role in astrocytes and stem cells, there is also work showing its requirement for normal oligodendroglial genesis. The 2003 paper by Stolt et al that was listed above, for example, showed that genetic knockout of Sox9 had delayed oligodendrogenesis. We were also not interpreting the Sox9 alone but in the context of the Olig2 data. While we have left the Sox9 data in place in the revised manuscript, we have now changed to description/interpretation to 'glial' (as opposed to oligodendroglial) – a modification that we believe more accurately describes it.

With regard to interpretation, as we allude to above in the general comments as well as the response to point 2, it is possible that the increase in OPCs is a product of two main factors: 1) that there are more NSCs / more NSC proliferation, 2) that the NSCs biased to choose the oligo lineage, and 3) that there is an effect on OPCs themselves that keeps OPCs in their proliferative state – possibilities that are non--mutually exclusive. To provide more focus and address the comment that the manuscript was previously somewhat diffuse, we now explain that we are addressing the first two, which occur at the NSC stage, and do not claim to directly address what happens after the OPC stage (although effects on the latter possibility still remain possible).

4) The conditional FoxG1--Cre experiment also demonstrates a clear increase in OPCs, however it is impossible to decipher whether these effects are specific to OPCs or are secondary to increases in neural stem cells. They should do a comprehensive timecourse in the FoxG1--Cre line and assess NSC proliferation, so that this figure shows both OPC and NSC data from the in vivo system. Also, if they want to claim an OPC effect, they should use an OPC--specific Cre, like Sox10--Cre or Olig2--Cre. Finally, they should also assess neurogenesis and astrocyte production in their existing lines and determine whether the changes they observed with these lineages in vitro are also observed in vivo, across a comprehensive timecourse.

Response: In revised Figures 2 and 3, we include greater quantitation of cells populations other than OPCs, and attempt to re-focus on the emphasis on the switch between neurogenesis and gliogenesis at the stem cell stage. We have not included OPC-specific drivers as part of this work due to reasons outlined above (we are not claiming to investigate a specific OPC effect, though do not excluded that possibility).

5) The link to Etv5 is convincing-----it's very clear that Cic directly represses Etv5. However, Etv5 function has been linked to astrocytes, not oligodendrocytes (see Fig. 5M of Li, et al. Neuron, 2012). This is a fundamental problem that is not at all consistent with their model.

Response: We are aware of the Neuron paper referenced that but we do not see the work of Li et al as fundamentally at odds with our interpretations. Li et al show that glial progenitor specification (for both astrocytes and oligodendrocytes) is adversely affected by decreased MEK signaling, and they link decreased MEK signaling to decreased ETV5. In the figure referenced where they overexpress Etv5 (Fig 5), the paper shows an impact on astrocytes but the data is moot on the impact to oligodendrocytes (they do not show data to indicate that it specifically does not affect oligodendrocytes). Because the authors have linked Etv5 function to astrocytes, to us, that does not mean that they propose that ETV5 function is exclusive to astrocytes and not any other cell type. Thus, we do not see the two bodies of work as being fundamentally problematic. In our opinion, this speaks to the novelty of our work.

In an effort to draw potential relevance to human disease oligodendroglioma, however, we recognize that we had presented a rather oligo-centric point of view and had emphasized oligodendrocytes throughout. We are certainly not proposing that Etv5 is not important in any other cell type, including astrocytes. Indeed, in our hands, we see that astros and oligos both express Etv5, and there are gains in both glial types with CIC loss (which we now show in revised figures 2 and 4). Previously, we had directed the narrow spotlight on the oligo aspects because we had found the changes were the greatest in that population. (Perhaps Li, et al, were using their own spotlight on a different cell type in gliogenesis. But the two are just complementary parts of the same reality – and it is the totality of all the spotlights of research into specific areas that will eventually illuminate the full picture.) Figuratively, instead of a directing a spotlight on oligos, the manuscript is now uses a floodlight shone on neural stem cells and their abilities to proliferate and choose fates of neurons, astrocytes, and oligodendrocytes.

For the in vitro experiments with Etv5, they may be showing a greater astrocyte impact than we detect due to differences in culture conditions and the strength of signaling through the RAS/MAPK pathway that may be present when Etv5 is or is not present. Nevertheless, overall, we do not see our data as fundamentally inconsistent with existing literature.

6) The epistasis test should be performed with a more robust and specific method of inhibition (like shRNAi) as the DN--ETV is likely to inhibit all ETS--family members. Along the same lines, when they overexpression ETV5 in fig5L, it has a nice effect on proliferation. What happens to cell fates? Does overexpression of ETV5 make OPCs? Astrocytes? NSCs? This set of experiments is very important and cellular outcomes should be fully analyzed.

Response: Although we have not repeated all our experiments with an RNAi approach, we have analyzed proliferation in the CIC null cells with siRNA against Etv4 and Etv5. With either Etv4 or Etv5 knockdown, proliferation of CIC null cells was decreased;; however the effect was greater with Etv5 siRNA than Etv4 siRNA. This data is now shown in a new Supplemental Figure S5 and is referred to in the main text.

In a new Figure 7, we show additional data indicating that in cultured cells, ETV5 overexpression during differentiation protocols phenocopies CIC loss. We display the data in newly formatted graphs that we believe are more intuitive to appreciate the functional relationship between CIC loss and Etv5 overexpression effects. Note: Although this is the only section where we do not provide directly matching experiments showing supporting *in vivo* data, we do provide *in vivo* data that generally speaks to the same point, showing that DN--ETV5 rescues the cell fate alterations induced by CIC loss (new Figure 3).

Reviewer #2

In this study, Ahmad et al. investigate the developmental role of the transcriptional repressor Cic in neural stem cells and in neural progenitors. In particular, the authors investigate the role of Cic in cell type specification and in cell fate regulation. The main findings demonstrate that Cic is expressed in both astrocyte and neuronal nuclei, but at lower levels in stem cells and in oligodendrocyte (OL) lineage cells. The authors generated a Cic floxed allele, and induced localized Cic deletion by electroporation of a Cre expression vector. Targeted Cic deletion in the forebrain increased proliferation and self--renewal of neural stem/progenitor cells. In vitro analysis of Cic--null NSCs demonstrated that Cic loss biases cells to the OL lineage, but maintains cells in an OPC state and delays their maturation. Finally, the authors demonstrate that these effects depend on de--repression of Etv5.

This is a potentially interesting study, revealing a new and early developmental role for a gene involved in oligodendroglioma formation. The information provides an appealing explanation for the oligodendroglial lineage bias arising from Cic loss in neural stem cells, beginning with loss of asymmetric division. However, there are many significant issues, the main one being that most of the study was performed in cultured cells and not in vivo. This significantly limits the impact and significance of the findings. Although the in vitro results are consistent with a cell specification role of Cic in neural cells, there are too many unanswered questions both in the in vivo and the in vitro part of the study, which make it largely incomplete – and unfocused. Neither the cell specification aspect, nor the role of Cic in OPC development (proliferation and differentiation) are investigated in detail. Finally, the role of Etv5 was not investigated in vivo.

Main issues

1. *Cic* expression in NSCs. This should be investigated more thoroughly both in NSCs and in their progenies in the SVZ, in the rostro--migratory stream (neuroblasts and NSCs) and in white matter OPCs derived from SVZ NSCs. More SVZ cell markers should be used, e.g. Nestin, Olig2, etc. Cell quantification (% of Cic+ cells expressing different markers) should also be obtained. Characterization shown in Figure 1 is superficial and not quantitative.

As suggested by this reviewer and the others, we have investigated more thoroughly CIC expression in several different cell populations. Previously, the comparisons were only between Sox2+, NeuN+, GFAP+, and Olig2+ cells. We now show additional quantitation with the following markers Aldh, CNPase, CC1, Tbr2, and Tbr1. This is shown in a revised Figure 1 and Supplemental Figure S1. Previously and also in the current revision, we have quantitated CIC expression using measurement of nuclear intensity of individual cells as a continuous variable rather than defining CIC expression as a binary state using an arbitrary cutoff (Cic+ and Cic-). This method is not only quantitative, but is more revealing to show graded differences in cell type specific expression than +/- alone. Perhaps what the reviewer was referring to with the comment that it was 'not quantitative' was that the figure was missing the bars indicating statistical significance between the groups. We have added the missing statistical analysis information. In case that is not what the reviewer desired, in a revised supplementary Figure 1, we also display what the % of Cic+ cells is in NeuN+, S100+, and Olig2+ cells using an arbitrary threshold.

2. The authors claim that SVZ analysis was performed in "adult" brain. However, the brains were analyzed at P21, which is not even young adult. Analysis should be performed at P90. How is Cic expression regulated in the adult SVZ as compared to the early postnatal SVZ (P7)? Do the levels change? Does Cic expression change in different progenitor cell types with age?

Although we have not performed an analysis at P90, as the reviewer suggested, we take the point raised about developmental timing. We have now quantitated using P56 (8 weeks) as the representative age for mature brain. At P56, mice are not undergoing any major neurodevelopmental processes and are fertile. P56 is also used by the Allen Brain Atlas as the age of analysis of adult mice. Although the number of cells of different types vary with age, we did not find any differences in the quantitated CIC expression intensity in cells of the same type across different ages. We have not displayed this comparison.

3. Figure 4. Analysis of gliogenesis. The characterization of the proliferation abnormalities induced by loss of Cic is rather complete, and it involves both in vitro and in vivo analysis. However, the effects on cell specification were exclusively investigated in cultured cells. This is a big gap in the study, as the effects on cell differentiation are truly novel.

Thank you for appreciating the novelty of our work. We now provide in vivo data in new Figures 2 and 3 showing that Cic deletion in vivo results in increased generation of stem cells, oligodendroglial lineage cells and astrocytes, but decreased production of neurons.

4. The authors show that Cic null cells display different lineage specification from control cells, in particular they express higher levels of Sox9 and Olig2 – i.e. Cic loss induces a pro--oligodendroglial program. This is only analyzed in culture and not in vivo. Differentiation defects were investigated by using typical OL lineage markers that identify different developmental stages. How does Cic ablation affect Sox2+ cells, since Sox2 has been reported to regulate the oligodendroglial lineage?

Response: We now document increased Sox9 and Olig2 in vivo in a new figure 4. Cic ablation increases Sox2+ cells – a finding that is also now illustrated in the new Fig 4 showing in vivo data.

5. Are Dcx neuroblasts not formed or apoptotic, and are differentiation abnormalities found in Cic--deficient cells in vivo? Do OPC differentiation abnormalities impact myelination? Given the lethality of Foxg1--targeted Cic ablation, an OPC--specific Cre driver is needed.'

For focus, we have intentionally kept the story to NSC proliferation and initial fate – and have not gone down each lineage with respect to their differentiation. Nonetheless, we detected no increase in apoptosis in Cic--ablated cells compared to control cells either in vitro or in vivo (documented in Supplemental figures S2 and S3). With regard to the question of OPC differentiation, we previously showed that the intensity of myelin staining was reduced in CIC knockout animals and the number of MBP+ cells was reduced in the in vitro differentiation assays. As described in the introduction to the rebuttal, however, we have decided to re--focus the revised manuscript on the stem cells proliferation and cell type specification aspects. Thus, we have not included experimental data using an OPC--specific Cre driver. (Of note, in a separate line of experiments, we are currently using PDGFRA--cre to directly address the effects of CIC loss in cells once they commit to the oligodendroglial lineage. Respectfully, we believe that should be a different paper though.)

6. It should be determined whether manipulation of Etv5 function in vivo can rescue the phenotype of the Cic--null mutant. The authors show that: i) Etv5 and 4 are upregulated after Cic knockdown, ii) DN--Etv5 lowers OPC and Olig2 cells, partially relieves differentiation block and reduces overall numbers of cells in Cic null cells, and iii) Etv5 overexpression causes NSC proliferation in culture. Would DN--Etv5 affect fate decisions in Cic--deficient NSCs? A link between these events should be established by demonstrating that knockdown of Etv5 in Cic--null mutant cells can rescue the phenotype in vivo.

Response: We have completed the series of ETV5 blockade analyses in vivo and show that much of the effect of CIC loss (not only the proliferation but also the altered cell fate decisions) can be rescued by DNETV5. This is now shown in the revised figure 3 and new Figure 7. We also include complementary data that speaks to this point in a new supplemental figure S5.

Reviewer #3

Ahmad et al. show in their manuscript “Capicua regulates neural progenitor cell proliferation and oligodendrocyte differentiation” an involvement of the Cic protein in the proliferation and differentiation properties of oligodendrocyte progenitor cells (OPCs) also implicating a possible role of Cic and OPCs in oligodendroglioma. By a newly generated Cic--flox mouse line, they could show in vivo and in vitro that deletion of Cic leads to increase in proliferation and in number not only of neural stem cells but also of OPCs through a switch of the division mode. Embryonic in vivo deletion lead to the development of a smaller forebrain as well as an increase in number of OPCs that they were stalled in their differentiation. At the end, the authors recognized EtV genes as important transcription factors mediating the OPCs differentiation process. Although interesting, the manuscript is in its current version pretty immature for publication in Nature Communications.

Major comments:

1. The authors claim that embryonically *Cic* is expressed in *Sox2*⁺ cells, however it is not only restricted to these cells. What other cells express *cic*? In the same line, at P21 *Cic* is low expressed in neural stem cells and high in differentiated cells. Are other progenitor cells in other areas, e.g. OPCs in the cortex and stem cells/transit amplifying progenitors in the dentate gyrus expressing *Cic*? To further support their hypothesis it would be nice to see that proliferating OPCs in the cortex do not express *Cic* in contrast to the non--proliferating OPCs that should express it.

Response: As outlined above, we have performed a much more extensive quantitation of *Cic* expression in different cell types/stages in the developing and adult cerebrum. Respectfully, we have not performed the suggested analysis of *CIC* expression in proliferating OPCs and non--proliferating OPCs in the cortex, however, as we have sought to increase the focus on describing and dissecting the mechanism behind the NSC proliferation and cell type specification phenotypes.

2. At P21, the authors use *GFAP* as a marker for astrocytes in the grey matter of the cerebral cortex. However, it is known that only reactive astrocytes in the grey matter (except in layer I) express *GFAP*-- It would be nice to see another more ubiquitous astrocytic marker, e.g. *S100b*, *Glast*, *Aldh1* etc. Similarly, using *Olig2* as an oligodendrocyte lineage marker is ok, but for the purposes of the manuscript it would be nice to separate between mature oligodendrocytes and OPCs by using specific markers, e.g. *NG2*, *PDGFRa* for OPCs, *CC1*, *Gstpi*, *CNPase* for mature and *MBP*, *MAG*, *PLP* for myelinating oligodendrocytes. This would be of great importance as the *Olig2*--staining was not very successful (unfortunately there is only one *Olig2*⁺ cell shown). An additional staining in areas with high numbers of cells of the oligodendrocyte lineage would be very useful.

Response:

While we recognize that *GFAP* does not mark 100% of astrocytes in the cortex, we respectfully disagree that it is only expressed in reactive astrocytes in the grey matter (except in layer I). In the literature, in my research, and in my professional role as a neuropathologist, I have seen *GFAP* present in astrocytes beyond layer 1, but the morphology changes and *GFAP* is further upregulated after they are reactive. Nevertheless, the point is well taken that additional markers for the various cell types would be helpful. Figure 1 depicting cell type specific difference in *CIC* expression has been substantially enhanced/revised. Instead of quantitating nuclear *CIC* intensity only in *Sox2*, *NeuN*, *GFAP*, and *Olig2* expressing cells;; we now assessed *Sox* (stem cells), *GFAP* and *Aldh1* (astrocytes), *PDGFRA* (OPCs), *CNPase* (early/pre--myelinating oligodendrocytes), *CC1* (mature oligodendrocytes), *Tbr2* (early born neurons), *Tbr1* (late born neurons), and *NeuN* (mature neurons). While we have not included as many oligodendrocyte lineage markers, particularly at the mature end of the spectrum, as the reviewer suggested to be performed (again, we have re--focused the paper overall more toward the stem cell / early lineage decisions and away from what happens subsequently), we hope that the new more detailed data and figure supplies enough information for the reviewers to feel confident in the overall conclusions. With respect to showing images in areas with a higher number of oligos, we have still used cortex but have shown an image with more than one oligodendrocyte. We chose this approach rather than showing, for example, a white matter tract with many oligodendrocytes, as we think it is easier for the reader to appreciate that the oligodendrocyte nuclei (despite being few in number in the image) have less intense *Cic* staining than neighboring non--

oligodendrocyte nuclei. In contrast, when a white matter tract is photographed, because most of the cells have less intense staining, it is more difficult to appreciate the differences. We hope that the new representative images are satisfactory.

3. The experiments with Cic deletion have to be described and analyzed more in detail. For example, in the in utero electroporated animals and in the ex-vivo transfected cells: -- What is the identity of cells that were electroporated/transfected? What was the transfection rate? What is the progeny of the electroporated cells (longer survival times would be nice)? Do only the electroporated/transfected cells (of which identity?) proliferate more or also the neighboring non-electroporated cells? This would be of importance to identify any cell non-autonomous effects. In the figure 3A it is very hard to see anything;; higher power images would be extremely helpful.

Response: With the IUE methods here and in previous work, we target only cells in the VZ;; very short term experiments (e.g. 6--8 hours with constructs such as pCIG2--Cre) indicate that the targeted cells are Sox2+ cells stem cells. We have not shown this data, as IUE is an established technique. Also, with this technique, the 'transfection efficiency' per se is not something that is readily quantitated, but the technique with the constructs we use informs us of which cells we should follow and count.

For the ex-vivo transfected cells, we transfected stem cell cultures (which in our hands are virtually all Sox2+ cells) and typically achieve a 40--50% efficiency. However for our assays, we selected for green cells so all the cells in our plasmid-transfected assays represent the desired cells.

With respect to the analyses themselves and the request for more phenotypic characterization, we not only supply information on the quantitation of the lineages that we sought to direct the cells towards (Fig 5) but also show the other lineages as well under each of the conditions (new supplemental figure S3). We also document that there is indeed a non-cell autonomous effect, at least for proliferation (supplemental Fig S6). While this is now mentioned/documentated, it is not an area that we feel is necessary to pursue that is central to the message of the manuscript (where we focus on cell autonomous effects of CIC on proliferation and lineage and the underlying mechanism of de-repressed ETV5).

4. In the in vitro experiments with the NSC-media, when the increase in Olig2-positive cells could be observed, it would be important to separate between OPCs, mature and myelinating oligodendrocytes. To further highlight the implication, do the authors also observe higher numbers of Olig2-positive cells in the neurospheres?

Response: We do observe higher numbers of Olig2+ cells in cells that are grown in stem cell conditions (Fig. 5A,A'). In these experiments, the cells were examined as adherent cells (on laminin instead of neurospheres) in order to facilitate quantitation. Although we did see an increase in Olig2+ cells, we did not detect any mature or myelinating oligodendrocytes when the cells were in stem cell conditions. We have not shown the latter negative data, but we do now mention it in the text.

5. The results of the cell cultures with the different media are very interesting and the decrease in neuronal and astrocytic markers under the respective culture conditions nice. However, it would be nice to know what the identity of the other cells is. Do the cells become oligodendrocytes or do they remain NSCs? In the same line, further

markers for astrocytes are needed as GFAP is a marker for reactive astrocytes. It could namely be that the observed effects are the result of (lack of) activation in these cells. Further, are the numbers of cells and their proliferation also changing? One would expect that subsets of NSC and/or OPCs (would be nice to know which one) are proliferating more while the ones primed to differentiate still do so. That would lead to a decrease in the proportion of neurons/astrocytes although the numbers remain constant.

Response: This data is now presented in a new Figure 6. We do see a small number of Olig2+ cells that are still proliferating at the end of the various courses of differentiation, but the differences are not significant between control and CIC null cultures. When we quantitate absolute numbers of cells (not only percentages), we see fewer Tuj1+ cells in the neuronal condition for Cic null cultures and fewer Gfap+ cells in the astrocytic condition for Cic null cultures. Thus, we conclude that the differences are at least in part due to decreased responsiveness to those differentiation signals for the specific cell type, and not only due to a persistent population of cycling cells 'diluting out' the targeted cell type. With respect to the identity of the other cells, we also show this in the new Figure 6. In the neuronal condition, of the Tuj1 negative cells, they mostly remain stem cells (Sox2+) with a smaller number choosing the oligo lineage (Olig2+). We include discussion of these findings also in the text.

For the points that we are making, we feel that examination of GFAP is adequate and that additional distinction between reactive and non--reactive astrocytes is not critical (would be unlikely to fundamentally change the story). We could perform such analysis, however, if the reviewer and editorial team think otherwise.

Another important issue would be how far the cells of the oligodendrocyte lineage are differentiating in the B27/TIT conditions. Do the cells remain in the progenitor state or do they differentiate?

Response: In the oligo differentiation conditions, we do eventually have some cells that express MBP or other myelinating markers at the RNA and protein level. MBP, for instance, is shown in Fig. 5E. Although the differentiation is far from complete in a majority of cells, we observe that there are a greater percentage of CIC null cells that are in progenitor stages and smaller percentage that express mature markers compared to control cultures. As noted previously, however, we have tried to de--emphasized possible effects on oligodendrocyte differentiation beyond the stage of lineage selection, as we have tried to bring more focus to the stem cell stage.

6. What is the phenotype of the mutant mice in regard to astrocytes, neurons and oligodendrocyte numbers? This would be important to understand if all the changes in the forebrain are due to proliferation changes in the OPCs or as the result of fate switch. What is the reason of the lethality?

Response: The astrocytes, neurons, and oligo numbers in the FoxG--Cre experiments is now shown in a revised Fig 2. Examining the dorsal cortical tissue (2D,D'), neurons show a relative decrease whereas astrocytes and oligos are relatively increased. When taken in aggregate with the data from the short term IUE and the cultured cell directed differentiation experiments, we interpret this as supporting a fate switch. (We cannot, however, exclude a concurrent effect on OPC proliferation and do not make claims to have investigated that. The question of OPC proliferation was the focus of the previously co--submitted manuscript, and we respectfully will leave that to our colleagues.)

The reason for lethality is not entirely clear. The mutant pups are born in approximate mendelian ratios, and at birth cannot be visually distinguished from their wildtype littermates. Within the first few postnatal days, however, they are visible runts that progressively fail to thrive. We have not found other major anatomic defects in other organs and suspect that inefficient feeding or other behaviors that may be related to impaired neurologic function may be related to their progressive decline after birth and their inability to survive past weaning. This, however, remains speculative. We have included statement to this effect in the main text.

7. In figure 5L they do not show any effects of Etv5 on proliferation as they claim in the text (line 315). The effects in cell numbers could also be due to survival differences and not due to changes in proliferation.

Response: We had previously described data on increased cell number as evidence of proliferation. Thank you for pointing out that inaccuracy. We have now performed both in vivo and in vitro experiments with Etv5 overexpression and have used EdU incorporation as the readout for these experiments (see new Fig. 7). Based on this, we confidently conclude that increased Etv5 is sufficient to drive proliferation both in vivo and in vitro.

We have not detected any increase in cell death in CIC null cultures compared to control or in CIC null electroporated cells (either by trypan blue or by activated caspase 3), so had previously concluded that the major increases in cell number are due predominantly to the proliferative effects that we show and not due to any significant difference in survival/death. Caspase 3 data on the CIC null and wildtype cells is now supplied as supplemental information in Supplemental figure S3. Because of this, we did not further pursue the evaluation of survival and death with the Etv5 overexpression experiments and only assessed the proliferation (which we document with Edu proliferation counts).

Minor comments

1. The numbers on line 275/276 do not correlate with the numbers in Fig, 4K. Similar on line 276/277 with fig. 4L. I assume that in the one case is proportion of all cells, in the other of Olig2--positive cells. Please clarify in the text.

Response: We have tried to clarify throughout by now using the formula for calculation in the figures and with changes to wording in the text. Yes, the one had referred to percentage of all cells. The other to the percentage of Olig2+ cells.

2. Please clarify what cell culture conditions were used for the ETV4/5 experiments.

Response: We have clarified that it is the oligodendrocyte culture condition.

3. Do oligodendroglioma also express higher levels of ETVs as the results would implicate?

Response: Yes, oligodendrogliomas also express higher levels of ETVs. This has been reported in existing literature, (Padul, et al, 2015 in the references that we had cited), and is also evident from TCGA data which we show in Supplemental Fig S5.

Reviewers' comments:

Reviewer #2 (Remarks to the Author):

The paper by Ahmad et al. has been resubmitted after complete revisions and reorganization of focus, text and figures. The authors have now focused the study exclusively on neural stem cells, in particular on the role of CIC in stem cell proliferation and lineage specification. The previous point of a role of CIC in OPC differentiation has now been removed. The authors have included a new focus, i.e. analysis in patient-derived oligodendroglioma cells (in vitro and in vivo) showing that CIC loss is also important in maintenance and proliferation of oligodendroglioma – and that CIC re-expression or Ets blockage reduce tumorigenicity.

The authors have also added a significant amount of new in vivo data, which are well integrated with the in vitro data in cultured cells. In vitro and in vivo data are consistent and support the major conclusions of the paper.

The new focus of the paper, together with the new data, has significantly strengthened the paper. The authors have also responded to all previous criticisms and addressed all issues either experimentally or by revising in the text.

Finally, the new results shown in Figure 10 demonstrating reduced tumorigenicity of ODG cells after either CIC re-expression or ETV5 blockade add to the significance and impact of the paper.

Reviewer #3 (Remarks to the Author):

Ahmad et al. show in their revised manuscript entitled "Capicua regulates neural stem cell proliferation and lineage specification through control of Ets factors" a role for the transcriptional repressor Cic in the proliferation and self-renewal of neural stem cells. Conditional deletion of Cic in the embryonic forebrain and in vitro leads to a decrease in neuronal fate and increase in glial lineage fate, both effects dependent on the de-repression of Ets transcription factors and specifically ETV5. Furthermore, as proof of principle, the authors used oligodendroglioma cell lines to further highlight the role of Cic and ETV5 in the tumorigenic potential.

The revised manuscript is strongly improved and it was a good move by the authors to restrict the paper on the stem cell proliferation and lineage specification. However, it would be nice to further address some more points:

- the authors jump between the panels in the different figures, this is pretty annoying. A rearrangement of the panels in figures fitting together, would be more appropriate and help a better understanding.
- It is still unclear what leads to the decrease in the cerebra size. Is it due to smaller grey matter, smaller corpus callosum or why? Fig S2A gives the opposite impression (thicker cortex). Is the layering normal? The small decrease in neurons would not explain the strong decrease in cortical size. Is the absolute number of neurons changed in the brain ?
- in the quantification of Olig2 cells in the P21 KO-brains around 17% are not PDGFRa or CNPase. What is the identity of these Olig2+ cells? Could they coexpress other markers like neuronal or stem cell markers?

- In the clonogenic assay, the higher number of spheres could also be the result of a stronger clumping of the cells (for whatever reason). As a proof of principle a single cell plating would be necessary.

- lines 300-302: 67.8% and 68.2% of the cells in the cic-wt are Tuj1+ and GFAP+ respectively. This makes (even without other cell types like cells of the oligodendrocyte lineage) more than 135%. Even if ones takes into account the SEM, the percentages are very high. Are there also double positive cells or how could this be explained?

- Fig. 8F/G: the controls in these two figures are very different (F: 30%, G: 50%). The decrease after DNETV5 expression in EdU+ cells to 30% (G) is very similar to the control in F. This effect could therefore also be the result of experimental variability.

- the difference between fig. 1J and S1c is not clear enough, please explain

- Fig. 1: highlight the area where the higher magnifications were taken from and add some scale bars in the inlays.

- line 193/194: "not" is missing.... The increased Sox2 fraction is not due to increased apoptosis...

RESPONSE TO REVIEWER COMMENTS

Thank you for your comments. We are happy that the revisions we had made were generally well received. As highlighted by the following comments from one reviewer:

“... added a significant amount of new in vivo data, which are well integrated with the in vitro data in cultured cells. In vitro and in vivo data are consistent and support the major conclusions of the paper. The new focus of the paper, together with the new data, has significantly strengthened the paper. The authors have also responded to all previous criticisms and addressed all issues either experimentally or by revising in the text.”

The other reviewer was also positive but had some remaining questions: “...revised manuscript is strongly improved and it was a good move by the authors to restrict the paper on the stem cell proliferation and lineage specification. However, it would be nice to further address some more points:”

Below, please find detailed responses to each of the queries raised by the latter reviewer.

1 - It is still unclear what leads to the decrease in the cerebra size. Is it due to smaller grey matter, smaller corpus callosum or why? Fig S2A gives the opposite impression (thicker cortex). Is the layering normal? The small decrease in neurons would not explain the strong decrease in cortical size. Is the absolute number of neurons changed in the brain?

We have revised Fig S2 and have added some additional description in the main text in lines 163-170 to provide more details as follows:

The overall decrease in the size of the cerebrum appears to be a global decrease in volume that affects gray matter as well as the white matter structures. The images shown previously had not been taken at the same anatomic level/location, and the slightly different locations resulted in the misleading appearance. To better illustrate the differences, we now provide images in a revised Suppl Fig 2 that are taken using the same exact same anatomic landmarks. The coronal sections showing the hippocampus and the dorsal cortex are now taken an anterior-posterior level through the anterior nucleus of the thalamus and the fields of cortex shown are directly superior to the hippocampus. The coronal images of corpus callosum and SVZ are taken at the level of the anterior commissure. In these images, the decreased brain size in the mutants is better appreciated.

With respect to the white matter, there is decreased thickness in white matter structures exemplified by the corpus callosum, which can be seen in Fig S2 panel B. Despite this decreased volume, there is a notable increased cellularity observed on H&E (that we had document as increased Olig2+ cells by immunofluorescence in the main figure 2).

In these cortical images, one can also see that there is a decrease in cortical thickness, however the relative lamination appears normal (no inversion of layers, no layers that are entirely missing). Of note, this finding is consistent with that of Lu, et al (Nat Genetics 2017) in figure 2 where they show that loss of Atxn1-Cic complexes resulted in reduced thickness in layers 2-4 and reduced neuronal counts of Satb2+ and Cux1+ cells. Because this was not our focus, and because the Zoghbi group already showed this, we did not pursue those specific further analyses.

With respect to numbers of neurons, we have quantitated the number of neurons per square mm in the dorsal cortex in the coronal sections that were taken from the level of the anterior commissure. As can be seen in the new Suppl Fig S2, the number of neurons per mm² in the cortex is not significantly different between mutants and controls. However because the volume of the cortex is smaller in mutants (while the neuronal density is the same), it is reasonable to conclude that the

absolute number of neurons in the cortex is lower in mutants compared to controls. We did not, however, perform a formal stereological assay through the brains. Of note, we do see an increase in total cellularity in the cortex (DAPI nuclei per mm²) in the mutant animals. These findings are consistent with the increase in glia that we have previously documented in the main Fig 2.

2 - In the quantification of Olig2 cells in the P21 KO-brains around 17% are not PDGFRa or CNPase. What is the identity of these Olig2+ cells? Could they coexpress other markers like neuronal or stem cell markers?

PDGFRa and CNPase were used to mark OPCs and later oligodendrocytes, respectively. It is well recognized, however, that these are not the only markers that can be co-expressed with Olig2 in a more stage-specific manner. For example, at the very earliest extreme, Olig2+ cells can co-express Sox2. As another example, O4 becomes expressed after the PDGFRa/OPC stage when cells are pre- or immature oligodendrocytes, and is then absent from the mature cells.

In a new Supplemental Figure S3 (referenced in lines 182-185), we show that in our brains, many of the Olig2+ but PDGFRa- and CNPase- negative cells consist of Olig2+Sox2+ cells. Sox2+Olig2+ cells account for 4.3% in control brains and 13.4% in the knockout brains. This finding is consistent with the other results throughout the manuscript suggesting that CIC loss both increases the NSC proliferation and biases their fate selection toward the oligodendroglial fate selection, resulting in an expansion of early-stage oligo lineage cells. We have not seen overlap of Olig2 with neuronal markers in our analyses.

3 - The authors jump between the panels in the different figures, this is pretty annoying. A rearrangement of the panels in figures fitting together, would be more appropriate and help a better understanding.

We appreciate the desire to make the reader experience as enjoyable. We were not sure, however, which figures seemed particularly annoying. We have put a lot of thought into the arrangement and feel that it is clear and efficient, despite occasionally asking readers to see two different figures. So we have not made any modifications to the figures other than what is described in the rest of this response (changes to Fig 1A to indicate areas for inlays; revised Suppl Fig S2; new Suppl Fig S3).

4 - In the clonogenic assay, the higher number of spheres could also be the result of a stronger clumping of the cells (for whatever reason). As a proof of principle a single cell plating would be necessary.

The reviewer has pointed out that when plating cells, it is possible that a higher number of spheres after plating a given number of cells could be the result of cell clumping. However the reviewer may have missed the portion of the methods section wherein we describe that we have plated in a semi-solid media (not in liquid media) for this particular assay. In the semi-solid media, the plated cells cannot move around and clump after plating as they might in liquid media. Repeating these series of experiments with single cell plating as second method is unlikely to change the conclusion that CIC-deficient cells more frequently have the capacity to self-renew and thus generate clones.

5 - Lines 300-302: 67.8% and 68.2% of the cells in the cic-wt are Tuj1+ and GFAP+ respectively. This makes (even without other cell types like cells of the oligodendrocyte lineage) more than 135%. Even if ones takes into account the SEM, the percentages are very high. Are there also double positive cells or how could this be explained?

The reviewer seems to have misunderstood the data contained in the previous lines 300-302 (now lines 318-322). The numbers to which the reviewer referred (67.8% Tuj1 and 68.2% GFAP) are from different experiments – one from a condition where the neural stem cells were grown in neuronal-promoting conditions and another in which the neural stem cells were grown in astrocytic promoting conditions. It is expected that when the uncommitted stem cells are cultured in a neuronal-promoting conditions, most become neurons. Similarly, it is expected that when the

uncommitted stem cells are cultured in astrocyte-lineage promoting conditions, most become astrocytes. (In contrast, there is no expectation that these two numbers should fundamentally add up to 100% or something below 100%. In point of fact, if the differentiation procedures are working, one would actually expect these two to add up to greater than 100, as they do.)

What we are showing and explaining here is that normally, in CIC-wt cells, most stem cell that are exposed to neuronal conditions become neurons (as expected), but that CIC-null cells do not as readily select the neuronal lineage when exposed to the same neuronal-inducing conditions. Likewise, most CIC-wt stem cells that are exposed to astrocytic conditions become astrocytes (as expected), but CIC-null cells do not as readily select the astrocytic lineage when exposed to the same astrocytic-inducing conditions.

We have re-read the section in question and still believe that the writing is clear. However we have made minor modifications in the text in that paragraph (lines 322-326 in the revised manuscript) to help underscore that point and avoid similar potential confusion by other readers.

6 - Fig. 8F/G: the controls in these two figures are very different (F: 30%, G:50%). The decrease after DNETV5 expression in EdU+ cells to 30% (G) is very similar to the control in F. This effect could therefore also be the result of experimental variability.

Figure 8, Panel F shows EdU incorporation in cultured CIC-wildtype NSCs with or without ETV5 overexpression. In contrast, Panel G shows EdU incorporation in CIC-null cells with or without DN-ETV5. These are two different types of cells in different experiments. There is no expectation that the respective controls (e.g. CIC wildtype NSCs at baseline for the experiment in Panel F, and the CIC-null cells at baseline in Panel G) in these experiments should have the same proliferation rate. Indeed based on the rest of our work, a higher proliferation rate in CIC-null cells is to be expected, and should not be taken as an indicator of extreme experimental variability.

With respect to the observation that proliferation of CIC-null cells after DNETV5 introduction goes down (decreased from 50% down to 30%), and this is similar to the proliferation increase of CIC-wildtype cells when they are made to overexpress ETV5 (proliferation goes up from about 30% in wildtype cells to about 50% in the cells overexpressing ETV5) actually underscores our conclusion made from the rest of the data that ETV5 is the major mediator of increased proliferation downstream of CIC loss. Here again, the result is actually supportive of our conclusions and does not suggest that the effect could be due to experimental variability.

7 - The difference between fig. 1J and S1c is not clear enough, please explain

We agree that the previous figures 1J and S1c were not very different at all. One shows the data as individual data points in a scatter plot (1J) while the other shows the data in bar graph format (S1C). Seeing that the two versions are very similar, we agree that it is not necessary to have the two and are now removing S1c bar graph, which does not contribute additional insight.

8 - Fig. 1: highlight the area where the higher magnifications were taken from and add some scale bars in the inlays.

We have now placed boxes indicating the areas that are shown at the right in grayscale for the particular color channel. Of note, these are not higher magnifications as suggested by the reviewer's comment, but are at the same magnification as the main panel. The boxes indicating the area represented in the main panel now make it obvious that the magnifications are the same. Thus to avoid clutter in the small grayscales, we are refraining from putting scale bars in each of the individual panes. If the editorial staff disagree and require the additional scale bars, please let us know and we will add them.

9 – Line 193/194: “not” is missing.... The increased Sox2 fraction is not due to increased apoptosis...

Indeed, the sentence was meant to read ‘ the increased Sox2 fraction is not due to increased apoptosis...” This has now been corrected (see current line 208). Thank you for your careful reading and for pointing out that error!

REVIEWERS' COMMENTS:

Reviewer #3 (Remarks to the Author):

The manuscript has very much improved and all the comments and concerns were addressed.